# Egg-laying and locomotory screens with *C. elegans* yield a nematode-selective small molecule stimulator of neurotransmitter release

Sean Harrington[1,2], Jessica J. Knox[2,3], Andrew R. Burns[2,3], Ken-Loon Choo[4], Aaron Au[2,5], Megan Kitner[6], Cecile Haeberli[7,8], Jacob Pyche[1,2], Cassandra D'Amata[2], Yong-Hyun Kim[4], Jonathan R. Volpatti[9,10], Maximillano Guiliani[2,5], Jamie Snider[2], Victoria Wong[2], Bruna M. Palmeira[11], Elizabeth M. Redman[11], Aditya S. Vaidya[12,13], John S. Gilleard [11], Igor Stagljar [2,3,14,15,16], Sean R. Cutler [12,13], Daniel Kulke[17,18,19], James J. Dowling[9,10], Christopher M. Yip [2,5], Jennifer Keiser[7,8], Inga Zasada[6], Mark Lautens[4] & Peter J. Roy [1,2,3✉]

Nematode parasites of humans, livestock and crops dramatically impact human health and welfare. Alarmingly, parasitic nematodes of animals have rapidly evolved resistance to anthelmintic drugs, and traditional nematicides that protect crops are facing increasing restrictions because of poor phylogenetic selectivity. Here, we exploit multiple motor outputs of the model nematode *C. elegans* towards nematicide discovery. This work yielded multiple compounds that selectively kill and/or immobilize diverse nematode parasites. We focus on one compound that induces violent convulsions and paralysis that we call nementin. We find that nementin stimulates neuronal dense core vesicle release, which in turn enhances cholinergic signaling. Consequently, nementin synergistically enhances the potency of widely-used non-selective acetylcholinesterase (AChE) inhibitors, but in a nematode-selective manner. Nementin therefore has the potential to reduce the environmental impact of toxic AChE inhibitors that are used to control nematode infections and infestations.

[1] Department of Pharmacology and Toxicology, University of Toronto, Toronto, Canada. [2] The Donnelly Centre for Cellular and Biomolecular Research, University of Toronto, Toronto, Canada. [3] Department of Molecular Genetics, University of Toronto, Toronto, Canada. [4] The Department of Chemistry, University of Toronto, Toronto, Canada. [5] Institute of Biomaterials and Biomedical Engineering, University of Toronto, Toronto, Canada. [6] USDA-ARS Horticultural Crops Research Laboratory, Corvallis, OR, USA. [7] Department of Medical Parasitology and Infection Biology, Swiss-Tropical and Public Health Institute, (Swiss TPH), Basel, Switzerland. [8] Faculty of Science, University of Basel, Basel, Switzerland. [9] Program in Developmental and Stem Cell Biology, The Hospital for Sick Children, Toronto, Canada. [10] Program in Genetics and Genome Biology, The Hospital for Sick Children, Toronto, Canada. [11] Department of Comparative Biology and Experimental Medicine, Host-Parasite Interactions (HPI) Program, Faculty of Veterinary Medicine, University of Calgary, Calgary, Canada. [12] Institute for Integrative Genome Biology, University of California, Riverside, CA, USA. [13] Department of Botany and Plant Sciences, University of California, Riverside, CA, USA. [14] Department of Biochemistry, University of Toronto, Toronto, Canada. [15] Mediterranean Institute for Life Sciences, Split, Croatia. [16] School of Medicine, University of Split, Split, Croatia. [17] Research Parasiticides, Bayer Animal Health GmbH, Monheim, Germany. [18] Present address: Department of Biomedical Sciences, Iowa State University, Ames, IA, USA. [19] Present address: Global Innovation, Boehringer Ingelheim Vetmedica GmbH, Ingelheim am Rhein, Germany. Preprint Servers: An earlier version of this manuscript is available on BioRxiv. https://doi.org/10.1101/2022.03.12.484074. It is made available under a CC-BY 4.0 International license. Classification: Biological Sciences, Genetics. ✉email: peter.roy@utoronto.ca

Parasitic nematodes are a scourge to humanity. Not only do helminths currently infect >1.5 billion people (WHO Soil Transmitted Helminth Fact Sheet, 2022) but nematode parasitism also leads to tens of billions of US dollars' worth of livestock losses annually[1]. Alarmingly, the rate of destruction is growing due to the evolution of anthelmintic resistance[2]. Plant-parasitic nematodes are even more destructive as they ruin over 125 billions of US dollars' worth of food crops annually[3]. Traditional nematicides that protect crops have been justifiably banned or severely restricted due to a lack of phylum selectivity[4], but these restrictions severely impact food security[5]. Compounding these issues, global food demand is expected to increase by 70% by the year 2050[6]. Hence, the development of new and selective nematicides is essential to our collective welfare.

As part of our effort to identify scaffolds with potential nematicidal/anthelmintic utility, we previously screened over 67,000 drug-like compounds for those that disrupt the life cycle of the free-living nematode *Caenorhabditis elegans*[7]. We called the resulting collection of 627 hits the worm-active (aka wactive) library. The wactive hits were re-screened against two parasitic nematodes and two vertebrate models, yielding 67 molecules belonging to 30 distinct small molecule scaffolds that selectively kill *C. elegans* and the parasitic nematodes *Cooperia oncophora* and *Haemonchus contortus*[7]. We showed that one of these scaffolds selectively kills nematodes via the inhibition of complex II (succinate dehydrogenase)[7].

Here, we present a motor-centric screening approach to identify candidate nematicides. Disrupting a parasite's motor control has repeatedly proven effective in mitigating nematode infection[8,9]. We therefore re-screened our wactive library using two successive behavioral assays of the free-living nematode *Caenorhabditis elegans*. We identified four molecules that fail to elicit phenotypes in off-target systems at concentrations that incapacitate multiple nematode species. We focused on one of these, called nementin, that induces *C. elegans* hyperactivity within seconds of exposure, followed by whole-body convulsions minutes later. Nementin and its analogs selectively incapacitate nematode parasites of plants and mammals. Chemical-genetic analyses reveal that nementin stimulates neuronal dense core vesicle release, which in turn, enhances synaptic vesicle release. This insight led to the finding that nementin enhances the paralytic effects of organophosphate and carbamate acetylcholinesterase inhibitors not only in *C. elegans*, but in nematode parasites of mammals and plants. We conclude that the nementin alkyl phenylpiperidine core scaffold is a nematode-selective nematicide lead that may also improve the selectivity of broad-acting pesticides.

## Results

**26 Compounds disrupt multiple *C. elegans* motor activities.** To identify small molecules that disrupt nematode neuromuscular activity, we designed a 96-well plate imaging system that measures the egg-laying rate of adult *C. elegans* hermaphrodites (Fig. 1a–c). The regulation of *C. elegans* egg-laying relies on well-characterized cholinergic, serotonergic, GABAergic and peptidergic circuits[10], and is therefore a paradigm to assess neuromuscular perturbation by small molecules. To identify egg-laying stimulators, we screened small molecules in the background of M9 worm buffer[7]. Animals incubated in M9-vehicle control wells lay 1.1 eggs per hour on average (Fig. 1d; Supplementary Data 1). To identify egg-laying inhibitors, we screened small molecules in the background of exogenous serotonin plus nicotine that stimulates animals to lay an average of 7.4 eggs per hour (Fig. 1d).

A screen of the 486 molecules from our worm-active library[7] revealed 29 stimulators and 29 inhibitors of egg-laying (Fig. 1e). We cross-referenced the list of acute egg-laying modulators to the list of 247 wactive molecules that are lethal in 6-day viability assays[7]. We found a 1.8-fold enrichment of lethality among the stimulators ($p < 0.001$) (Fig. 1f), suggesting that excessive neuromuscular modulation may lead to death. A 1.4-fold enrichment of lethality among the inhibitors was also found ($p < 0.05$), which is not unexpected given that egg-laying inhibition is certain to be a consequence of molecules that kill rapidly. A Tanimoto structural similarity cut-off of 0.55 reveals that the 58 egg-laying modulators can parsed into 37 distinct core scaffolds, consisting of 11 clusters and 26 singletons[11] (Fig. 1g). 10 of the 11 clusters are composed of molecules with the identical egg-laying phenotype, which validates the phenotypic assignments.

We reasoned that molecules that disrupt multiple motor circuits are more likely to disrupt the parasitic nematode lifecycle. We therefore surveyed the 58 egg-laying modulators for those that disrupt the normal sinusoidal locomotion of wild-type *C. elegans*. 26 of the 58 egg-laying modulators (45%) induce at least one locomotory phenotype such as convulsions, coiling, and paralysis (Fig. 2; Supplementary Movies 1–6;). The molecules that induce slow movement and paralysis are dominated (88%) by molecules that inhibit egg-laying (Fig. 2). This is consistent with the idea that the egg-laying inhibitor class may be enriched with acutely lethal molecules. By contrast, the molecules that induce convulsions, shaking, and coiling are dominated (80%) by those that enhance egg-laying (Fig. 2).

**Nementin-1 is effective against multiple parasitic nematodes.** To prioritize the 26 neuromodulatory molecules, we eliminated those from consideration that: (i) are cytotoxic to human HEK293 cells; (ii) induce developmental defects in zebrafish; (iii) have lackluster lethality against the five nematodes previously assayed; (iv) induce subtle locomotory phenotypes (reversal-defective), or (v) were closely related to the commercial nematicide fluopyram[7] (Fig. 2). This left worm-active (wact) molecules 10, 13, 15, 55, 120, 128, and 444 for further consideration (Fig. 3a).

We further assessed the activity of our seven hits against seven nematode species from three phylogenetic clades (Fig. 3b; Supplementary Data 2). Four of these seven species are parasites of mammals. We also analyzed (or reanalyzed) the molecules' activity against the non-target models *D. rerio* fish, HEK293 cells, and *Arabidopsis thaliana* plants (Fig. 3b; Supplementary Fig. 1). We prioritized wact-55 because it arguably demonstrates the best combination of broad nematode activity and selectivity. SciFinder-based literature searches failed to reveal prior annotation of nematicidal activity for wact-55's alkyl phenylpiperidine core scaffold[12].

A detailed temporal analysis showed that within 90 s of exposure of 60 μM wact-55, *C. elegans* exhibits a hyperactive nose movement phenotype (Fig. 3c). The hyperactive phenotype dissipates at the expense of spastic convulsions over the course of 2 h (Fig. 3c). By 24 h, 100% of the adult animals incubated in wact-55 are dead (non-moving and/or disintegrating; three trials, $n = 18$ animals per trial). For reasons outlined below, we renamed the wact-55 molecule 'nementin-1' (nematode enhancer of neurotransmission-1).

To determine whether nementin-1 might have a canonical mechanism-of-action, we tested whether any of the *C. elegans* mutant strains that resist the effects of seven commercially available anthelmintics also resist the lethal effects of nementin-1. All drug-resistant strains remain sensitive to nementin-1 (Supplementary Fig. 2). Furthermore, we previously showed that *C. elegans* cannot be easily mutated to resist nementin-1's lethality; 290,000 randomly mutated genomes failed to yield wact-55-resistant mutants[7]. These results suggest that: (i) nementin-1 does not share a mechanism-of-action with the tested anthelmintics, (ii) nementin-1's mechanism-of-action may not be limited to a single protein target, and, (iii) genetic resistance to

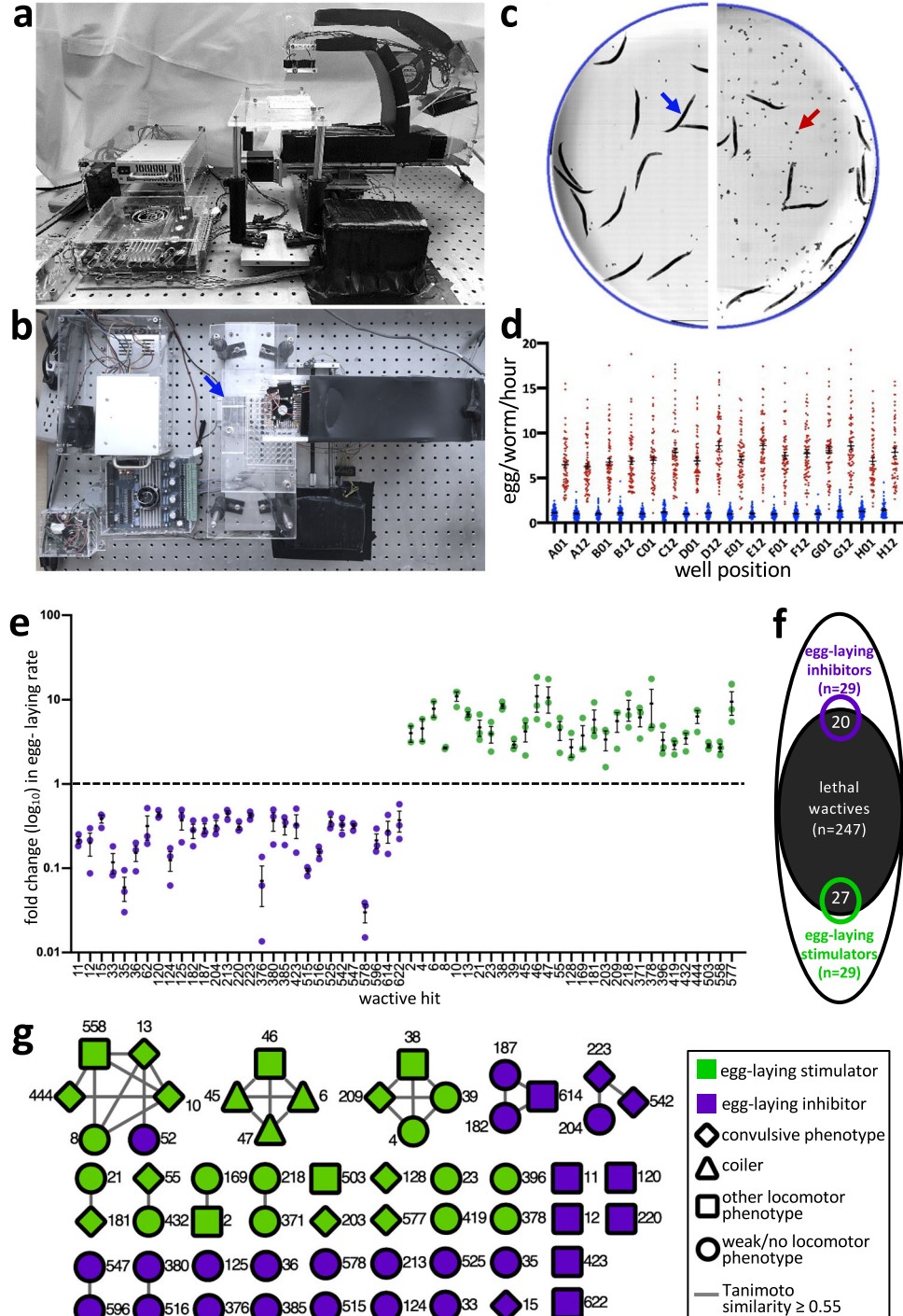

**Fig. 1 The identification of 58 small molecules that modulate the *C. elegans* nervous system. a, b** Two perspectives of the '2020 Imager' showing one 96-well plate in the imager holder (arrow). **c** Representative half-well images of vehicle controls in the basal (M9-only) background (left) and the 'stimulatory cocktail' of 12.3 mM serotonin creatine sulfate and 7.7 mM nicotine (right) after 1 h. A blue arrow points to an adult hermaphrodite and a red arrow points to a pair of embryos. **d** The distribution of counts of eggs/worm/hour for the basal background (blue) and the stimulatory background (red) from the indicated well-coordinates over 22 experimental replicates (96-well plates). The mean and SEM is shown, $n = 66$ biologically independent samples with ~20 animals per data point. **e** A screen of the worm active (wactive) library yields 29 inhibitors (purple) and 29 stimulators (green) of the *C. elegans* egg-laying rate. Mean Egl rate and SEM is shown from $n = 3$ independent biological replicates, conducted in technical triplicate with ~20 animals per technical replicate. **f** Venn diagram showing the overlap of the wactive molecules that are lethal (at 60 μM) with those that modulate egg-laying among the 486 molecules assayed in both assays. **g** A structure similarity network of the 58 egg-laying modulators. Each node represents a numbered wactive molecule. Connecting lines indicate shared structural similarity. Additional information is provided in Fig. 2. Primary data for this figure can be found in Supplementary Data 1.

| | wactive | Chembridge ID a | egg-laying phenotype b | locomotor phenotype c | | | | | | | larval lethal assay d | | | | | HEK293 e | Danio rerio f |
|---|---|---|---|---|---|---|---|---|---|---|---|---|---|---|---|---|---|
| | | | | shrinker | shaker | coiler | jerky-Unc | reversal-defective | paralyzed | slow | C. elegans | C. briggsae | P. pacificus | C. oncophora | H. contortus | | |
| **Convulsive (primarily)** | | | | | | | | | | | | | | | | | |
| 1 | wact-10 | 5905384 | Egl-S | 30 | 60 | n | n | n | n | n | 8 | 30 | 30 | 4 | 60 | n | n |
| 2 | wact-13 | 7632721 | Egl-S | 30 | n | n | 30 | 60 | n | n | 8 | 30 | 30 | 8 | 60 | n | n |
| 3 | wact-15 | 7687977 | Egl-I | 30 | n | n | 30 | 30 | n | n | 8 | 60 | 60 | 8 | 60 | n | n |
| 4 | wact-55 | 5688274 | Egl-S | 60 | 60 | n | n | n | 30 | n | 8 | 30 | 30 | 8 | 8 | n | n |
| 5 | wact-128 | 7971816 | Egl-S | 30 | n | n | n | n | n | n | 30 | 30 | 30 | 4 | 60 | n | n |
| 6 | wact-181 | 9032631 | Egl-S | 30 | >60 | n | n | n | n | n | 30 | 30 | 8 | 15 | 60 | 30 | dead |
| 7 | wact-209 | 7153658 | Egl-S | 30 | 60 | n | n | n | n | n | 30 | 30 | 30 | 15 | 60 | 60 | n |
| 8 | wact-223 | 7638283 | Egl-I | 60 | n | n | n | n | n | n | n | n | n | 60 | 60 | 60 | n |
| 9 | wact-542 | 7781713 | Egl-I | 30 | n | n | n | n | n | n | n | n | 60 | 15 | 60 | n | dev |
| **Shaker (primarily)** | | | | | | | | | | | | | | | | | |
| 10 | wact-203 | 9062286 | Egl-S | >60 | 60 | n | 60 | n | n | n | 30 | 30 | 30 | 15 | 60 | n | dev |
| 11 | wact-444 | 5910902 | Egl-S | n | 30 | n | n | n | n | n | 30 | 30 | 30 | 15 | 60 | n | n |
| 12 | wact-577 | 9039023 | Egl-S | n | 60 | n | n | n | 60 | n | 60 | 60 | 60 | 60 | 60 | n | n |
| **Coiler (primarily)** | | | | | | | | | | | | | | | | | |
| 13 | wact-6 | 5419367 | Egl-S | n | n | 30 | n | n | 30 | n | 30 | 30 | 30 | >60 | >60 | 30 | card |
| 14 | wact-45 | 5426270 | Egl-S | n | n | 30 | 60 | n | 60 | n | n | n | n | 60 | >60 | n | n |
| 15 | wact-47 | 5427063 | Egl-S | n | n | 60 | >60 | n | >60 | n | 30 | n | n | 60 | >60 | 30 | card |
| **Reversal Defective (primarily)** | | | | | | | | | | | | | | | | | |
| 16 | wact-38 | 5356411 | Egl-S | n | n | n | n | 30 | n | n | 30 | 30 | 30 | 8 | 60 | n | n |
| 17 | wact-558 | 9008591 | Egl-S | n | n | n | n | 60 | n | 30 | 8 | 30 | 30 | 60 | 60 | n | n |
| **Jerky-Unc (primarily)** | | | | | | | | | | | | | | | | | |
| 18 | wact-46 | 5426998 | Egl-S | n | n | n | 30 | n | >60 | n | n | 60 | n | 8 | >60 | n | card |
| 19 | wact-120 | 7889289 | Egl-I | n | n | n | 60 | n | n | n | 30 | 30 | 30 | 4 | 60 | n | n |
| **Paralysis** | | | | | | | | | | | | | | | | | |
| 20 | wact-11 | 6222549 | Egl-I | n | n | n | n | n | 30 | n | 8 | 8 | 8 | 8 | 60 | n | n |
| 21 | wact-12 | 7003409 | Egl-I | n | n | n | n | n | n | n | 8 | 30 | 8 | 8 | 60 | n | n |
| 22 | wact-220 | 5784085 | Egl-I | n | n | n | n | n | 30 | n | 8 | n | 30 | >60 | 60 | n | dev |
| **Slow** | | | | | | | | | | | | | | | | | |
| 23 | wact-423 | 5547023 | Egl-I | n | n | n | n | n | n | 30 | 60 | n | 60 | 8 | 60 | 30 | n |
| 24 | wact-503 | 7568929 | Egl-S | n | n | n | n | n | n | 30 | 60 | 60 | 60 | 60 | 60 | 30 | n |
| 25 | wact-614 | 9039813 | Egl-I | n | n | n | n | n | n | 30 | 8 | 30 | 60 | >60 | >60 | 30 | dead |
| 26 | wact-622 | 5373894 | Egl-I | n | n | n | n | n | n | 30 | n | n | n | 60 | >60 | 30 | n |
| **Weak or No Locomotory Phenotype** | | | | | | | | | | | | | | | | | |
| 27 | wact-2 | 5185411 | Egl-S | n | n | n | n | n | n | 60 | 8 | 8 | 8 | 8 | >60 | 60 | dev |
| 28 | wact-4 | 5352487 | Egl-S | >60 | n | n | n | n | n | n | 30 | 30 | 30 | 30 | 60 | n | n |
| 29 | wact-8 | 5652977 | Egl-S | n | n | n | n | >60 | n | n | 30 | 8 | 8 | 60 | >60 | 30 | dead |
| 30 | wact-21 | 5129511 | Egl-S | n | n | n | >60 | n | n | n | 30 | n | n | 8 | 60 | n | n |
| 31 | wact-23 | 5156707 | Egl-I | n | n | n | n | >60 | n | n | 30 | 30 | 8 | 60 | 60 | 30 | dead |
| 32 | wact-33 | 5308651 | Egl-I | n | n | n | n | n | n | n | 30 | 30 | 30 | 8 | >60 | 30 | n |
| 33 | wact-35 | 5344384 | Egl-I | n | n | n | 60 | n | n | n | 30 | 60 | 60 | 8 | 60 | 30 | n |
| 34 | wact-36 | 5347942 | Egl-I | n | n | n | n | n | n | n | n | n | n | 8 | 60 | n | n |
| 35 | wact-39 | 5357418 | Egl-S | n | n | n | n | n | n | n | 30 | n | 30 | 8 | 60 | n | n |
| 36 | wact-62 | 6269333 | Egl-I | depleted | | | | | | | 30 | 30 | 60 | 8 | 60 | n | n |
| 37 | wact-124 | 7943845 | Egl-I | n | n | n | n | n | n | >60 | 30 | 30 | 30 | 8 | 60 | 60 | card |
| 38 | wact-125 | 7949759 | Egl-I | n | n | n | n | n | n | n | 30 | 60 | 8 | 60 | >60 | n | n |
| 39 | wact-169 | 9024879 | Egl-S | n | n | n | n | n | n | n | 8 | 30 | 8 | >60 | >60 | n | n |
| 40 | wact-182 | 9033974 | Egl-I | n | n | n | n | 60 | n | n | 30 | 60 | n | n | n | 60 | n |
| 41 | wact-187 | 9035461 | Egl-I | n | n | n | n | n | n | n | <<30 | 60 | n | n | n | 60 | n |
| 42 | wact-204 | 9063096 | Egl-I | n | n | n | n | >60 | n | n | 30 | 60 | 30 | >>60 | >60 | 30 | dead |
| 43 | wact-213 | 5107544 | Egl-I | n | n | n | n | n | n | n | 60 | n | n | >60 | >60 | n | n |
| 44 | wact-218 | 5379978 | Egl-S | n | n | n | >60 | n | n | n | 8 | 30 | 30 | 15 | 60 | 30 | n |
| 45 | wact-371 | 5100784 | Egl-S | n | n | n | n | n | n | n | 8 | 60 | 60 | 8 | 60 | 30 | n |
| 46 | wact-376 | 5192203 | Egl-I | depleted | | | | | | | 60 | 30 | 30 | 8 | 60 | n | dead |
| 47 | wact-378 | 5192696 | Egl-S | n | n | n | n | n | n | 60 | 30 | 60 | 60 | 8 | 60 | n | n |
| 48 | wact-380 | 5218876 | Egl-I | n | n | n | n | n | n | n | n | n | 30 | >60 | n | n | card |
| 49 | wact-385 | 5238652 | Egl-I | n | n | n | n | >60 | n | n | 30 | 60 | 60 | 8 | 60 | 30 | dead |
| 50 | wact-396 | 5322542 | Egl-S | n | n | n | n | n | n | n | 60 | 60 | 60 | 8 | 60 | n | n |
| 51 | wact-419 | 5469460 | Egl-S | >60 | >60 | n | n | n | n | n | 60 | 60 | 60 | 60 | 60 | n | n |
| 52 | wact-432 | 5689365 | Egl-S | n | n | n | n | n | n | n | 60 | 60 | 30 | 8 | 60 | n | n |
| 53 | wact-515 | 7653692 | Egl-I | n | n | n | >60 | n | n | n | n | n | 60 | 60 | 60 | 30 | n |
| 54 | wact-516 | 7664300 | Egl-I | >60 | n | n | n | n | n | n | >60 | 60 | 30 | 4 | 60 | n | card |
| 55 | wact-525 | 7705966 | Egl-I | n | n | n | n | n | n | n | n | n | 60 | 60 | 60 | 60 | n |
| 56 | wact-547 | 7877602 | Egl-I | n | n | n | n | n | n | n | n | n | 60 | >60 | >60 | 30 | dead |
| 57 | wact-578 | 9040453 | Egl-I | n | n | n | n | n | n | n | 30 | n | n | 8 | 8 | n | dead |
| 58 | wact-596 | 5649594 | Egl-I | n | n | n | n | n | n | n | 30 | 30 | 30 | >60 | 60 | n | dead |

nementin-1 may be difficult to achieve in the field (which is a hypothesis that remains to be tested). All of these properties make nementin-1 an attractive hit for further investigation.

**Nementin convulsions are dependent on dense core vesicle release.** We reasoned that mutations in the pathway targeted by nementin-1 might phenocopy the motor defects induced by the molecule and provide mechanistic insight. A survey of the literature yielded eight mutants that share at least some of nementin-1's phenotypes, including *acr-2, unc-2, unc-43, unc-58, unc-93, sup-9, sup-10,* and *twk-18*[13–21] (Supplementary Table 1). We tested the hypothesis that nementin-1 disrupts the pathway disrupted by each of these mutants. We did this by asking whether known suppressors of the phenocopying mutants can suppress nementin-1's

**Fig. 2 *C. elegans* locomotor phenotype and phylogenetic activity profile of identified egg-laying modulators. a** The Chembridge Inc. identification (ID) numbers are indicated. **b** Egg-laying phenotype; Egl-S egg-laying stimulators; Egl-I egg-laying inhibitors. **c** The observed acute motor phenotypes are indicated. The concentration at which strong and moderate phenotypes appear are indicated in bright green. The concentration at which weaker phenotypes appear are indicated in light gray. 'n' indicates no phenotype was observed. **d** Larval lethality phenotypes; the previously reported results[7] of liquid-based larval lethal assays (for *C. elegans*, *C. briggsae*, and *P. pacificus*) are shown. The lowest concentration at which 100% of the larvae die/arrest is shown. Colors highlight relatively potent activity. **e** HEK293 cell proliferation summary; compounds that reduce HEK293 proliferation below one standard of deviation from the mean at 30 or 60 μM as previously reported are indicated as purple '30' or '60', respectively (see[7] for details). **f** *Danio rerio* (zebrafish) developmental defect summary; molecules that induce cardiac defects (card), developmental defects (dev) or death (dead) at a concentration of of 10 μM as previously reported[7] are summarized here. Throughout the figure, 'n' reports no lethality or phenotype observed. Primary data for this figure can be found in Supplementary Data 1.

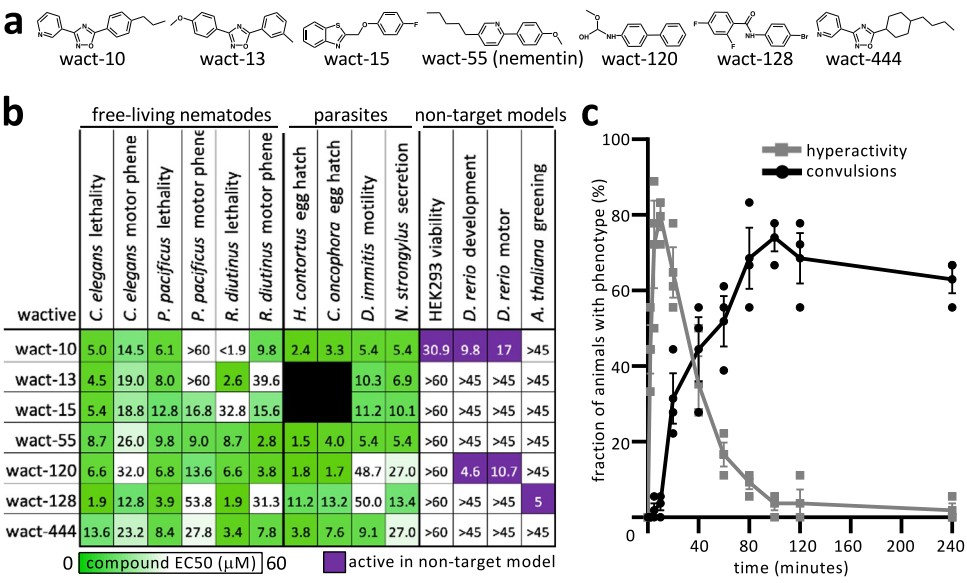

**Fig. 3 The motor-centric approach yields convulsion-inducing nementin-1. a** Structures of the seven prioritized neuromodulators. **b** Bioactivity of the seven prioritized molecules. Data are EC50s (details of assays are provided in the methods) except for the *A. thaliana* greening assay, which is the lowest concentration that yields a discernible difference from control in at least two of the three biological replicates tested (*n* = 3). **c** A time-course analysis of 60 μM nementin-1-induced locomotory phenotypes. Data are the mean of biological triplicate measurement of 18 animals per each time point over three biologically independent replicates (*n* = 3). The SEM is shown. Primary data for this figure can be found in Supplementary Data 2.

effects (see Supplementary Table 1). Of the potential suppressing mutants tested, only the *unc-43* gain-of-function (GF) mutant resisted nementin-1's convulsions (Fig. 4a). We also found the converse to be true; *unc-43* reduction-of-function (RF) alleles enhance nementin-1 convulsions (Fig. 4a; Supplementary Data 3). Remarkably, the interaction between the *unc-43* GF and nementin-1 is mutual in that nementin-1 can rescue the paralysis of *unc-43* GF (Fig. 4b). This observation argues against the idea that the *unc-43* GF mutant alters nementin-1 absorption or metabolism. These results suggest that nementin-1 may be disrupting UNC-43 pathway activity.

UNC-43 is the *C. elegans* ortholog of CaMKII (calcium/calmodulin-dependent protein kinase II) and is a key negative regulator of dense core vesicle (DCV) release in *C. elegans* neurons[14,22] (Fig. 4c). In animals with reduced UNC-43 function, neuronal DCV content is released in excess and is likely responsible for the mutant's convulsive and constitutive egg-laying phenotypes[13,22]. In *unc-43* GF animals, DCVs accumulate, and their limited release is likely responsible for the mutant's severely lethargic locomotion[22].

To examine DCV behavior in response to nementin-1, we exploited the neuropeptide reporter (INS-22::GFP) that is packaged into DCVs in cholinergic motor neurons[22]. Like *unc-43* RF mutants[22], worms treated with either nementin-1 or a second analog called nementin-12 have reduced axonal INS-22::GFP signal (*p* < 0.05) (Fig. 4d–f, h; Supplementary Fig. 3c, d;

Supplementary Data 3). Coincidentally, INS-22::GFP accumulates in the pseudocoelomic fluid-scavenging cells, called coelomocytes, of nementin-treated animals (Fig. 4g).

We reasoned that if nementin disrupts motor behavior by stimulating DCV release, then disruption of UNC-31 (calcium-dependent activator protein for secretion (CAPS)), which is required for DCV release[22,23], or UNC-64 syntaxin, which is required for all vesical fusions[24,25], should suppress the nementin-1-driven locomotory defects. Indeed, we find that both the weaker temperature-sensitive *unc-31(e169)*, the *unc-31(e928)* null, and the canonical *unc-64(e246)* RF mutants suppress the convulsions induced by nementin-1 (Fig. 4a). Together, these data indicate that nementin-induced DCV release is responsible for convulsions.

Collectively, these results beg the question of whether UNC-43 is likely to be the physiologically relevant target of nementin-1. Several observations argue against this. First, we note that the terminal phenotype of nementin-1 is death, yet presumptive null alleles of *unc-43* (*ce685* and *n1186*) are viable[14,22]. Second, if nementin-1 inhibits UNC-43, then nementin-1 should not be able to enhance the convulsion phenotype of *unc-43* null mutants. We find that nementin-1 does indeed enhance presumptive *unc-43* null mutants (*ce685* and *n1186*) (Fig. 4a). Third, if UNC-43 was nementin-1's sole physiological target, then *unc-43* GF mutations would have been identified in our genetic suppressor screens[7], but no resistant mutants arose. Finally, the *unc-43* GF mutant

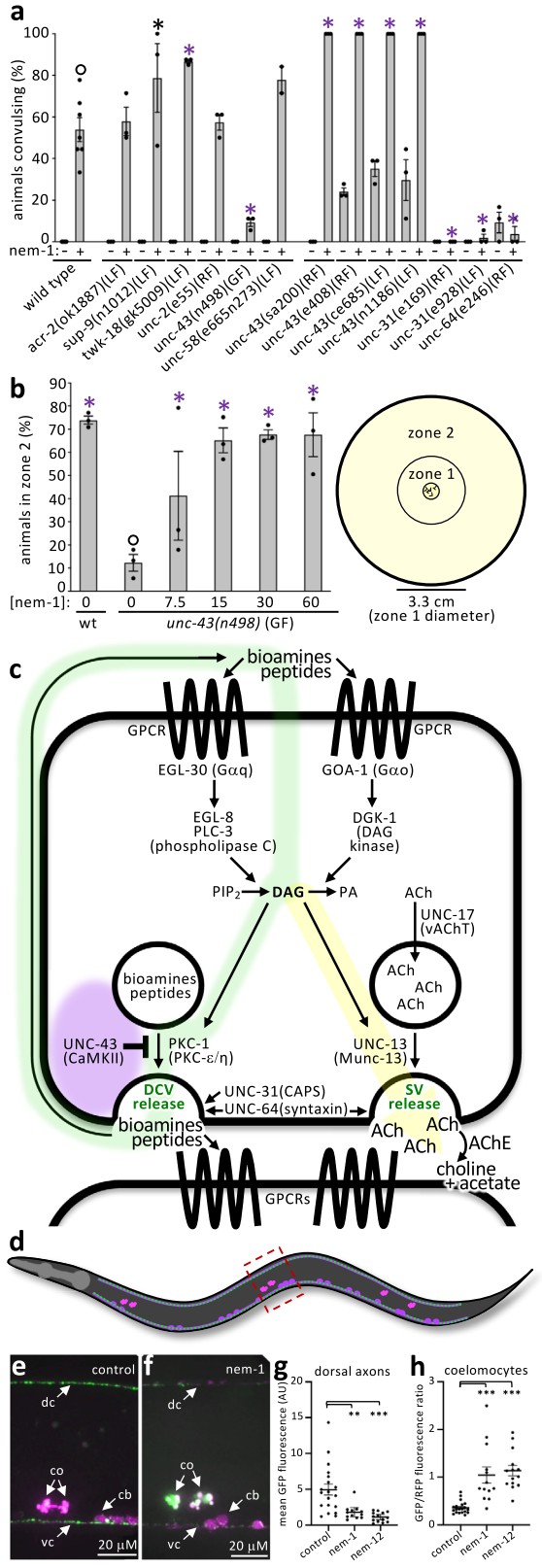

**Fig. 4 Nementin stimulates dense core vesicle release. a** Convulsions induced by nementin-1 (Nem-1) (60 µM) in the indicated genetic background after 80 min ($n = 3$ biologically independent samples scoring 18 animals per trial), with the exception of 60 µM wild-type nementin ($n = 7$ scoring 18 animals per sample). **b** The mean percentage of animals of the indicated genotype ($n = 3$ biologically independent trials with ~150 animals per trial) that locomote to zone 2 over 180 min after being placed in the center of a plate (see schematic on right). For A and B, black and purple asterisks represent $p < 0.05$ and $p < 0.001$, Chi-square test of independence with Bonferroni correction relative to control (open circle); the standard error of the mean (SEM) is shown. **c** A schematic of neuronal signaling pathways relevant to nementin activity. Purple glow indicates area of possible nementin action; the green glow indicates how DCV neurotransmitter cargo may activate $G\alpha_q$ and $G\alpha_o$ signaling pathways; the yellow glow indicates how the secondary messenger diacylglycerol (DAG) can also promote synaptic vesicle release. **d** Schematic of adult (strain KG4247) expressing INS-22::GFP (green dots; packaged into DCVs) and secreted mCherry (fuchsia) in cholinergic motor neurons (cell bodies are oval). mCherry is constitutively secreted and taken up by the coelomocytes (three pairs of pink objects in the animal). Box indicates the area shown in E and F. Anterior is the left and dorsal is up. **e, f** Images of the midbody region of control and 60 µM nementin-1-treated KG4247 animals after 4 h. dc, dorsal cord; vc, ventral cord; co, coelomocytes; cb, cell bodies. **g, h** Quantification of (**g**) the midbody dorsal cord fluorescence (relative to background tissues) and (**h**) mid-body coelomocytes (ratio of measured GFP/RFP). AU, arbitrary units., 1-way ANOVA with Dunnett correction for multiple comparisons; **$p < 0.01$; ***$p < 0.001$; SEM is shown. Other regions of the animal are quantified in Supplementary Fig. 3. Primary data for this figure can be found in Supplementary Data 3.

**Nementin-1 enhances cholinergic signaling.** We next investigated whether cholinergic signaling is also required for nementin-1 activity. We tested whether mutants of the UNC-17 vesicular acetylcholine transporter (VAChT), which loads synaptic vesicles (SVs) with acetylcholine, or mutants of UNC-13, which is specifically required for SV release[26,27], could suppress nementin-1-induced convulsions. Reduction-of-function mutants of UNC-17 and UNC-13 remained sensitive to nementin-1 (Fig. 5a; Supplementary Data 5). Incidentally, we found that nementin-1 could partially rescue the lethargic locomotion of *unc-17* RF mutants (Fig. 5b). These data show that despite cholinergic signaling being dispensable for nementin-1-induced convulsions, nementin-1 may nevertheless enhance cholinergic signaling.

We further investigated the idea that nementin-1 enhances cholinergic signaling. To do so, we used the acetylcholinesterase (AChE) inhibitor aldicarb, which is a well-characterized tool used to investigate perturbations of cholinergic signaling in *C. elegans*[28]. AChE catabolizes acetylcholine at the neuromuscular junction (NMJ) and its inhibition increases acetylcholine levels that in turn paralyzes the animal[29] (see Fig. 4c). We found that nementin-1 sensitizes animals to the paralytic effects of aldicarb (Fig. 5c), suggesting that nementin-1 stimulates acetylcholine release at the NMJ. The relationship between nementin-1 and aldicarb is synergistic in nature, yielding a zero-interaction potency (ZIP) δ-score of 31.1 with a δ-score of 64.8 over the most synergistic set of concentrations (ZIP scores > 10 are considered synergistic[30,31] (Fig. 5d, e)). Nementin-1 also enhances the paralysis induced by dichlorvos and trichlorfon, which are two other AChE inhibitors that are also commercial anthelmintics, by 12.7 and 18.7-fold, respectively (Fig. 5c). By contrast, nementin-1 fails to enhance the effects of an AChE-inhibitor on the fruit fly *D. melanogaster* (Fig. 5c). These data provide additional support for the idea that nementin-1 enhances cholinergic signaling and does so in a nematode-selective manner.

(and *unc-31* and *unc-64* RF mutants) remain sensitive to the lethal effects of nementin-1 (Supplementary Fig. 4; Supplementary Data 4). Together, these data argue against the idea that UNC-43 is nementin-1's sole physiologically relevant target. Instead, the similarities between nementin-1-induced phenotypes and the *unc-43* RF mutants suggest that the compound targets a component(s) that acts with UNC-43 (Fig. 4c, purple glow).

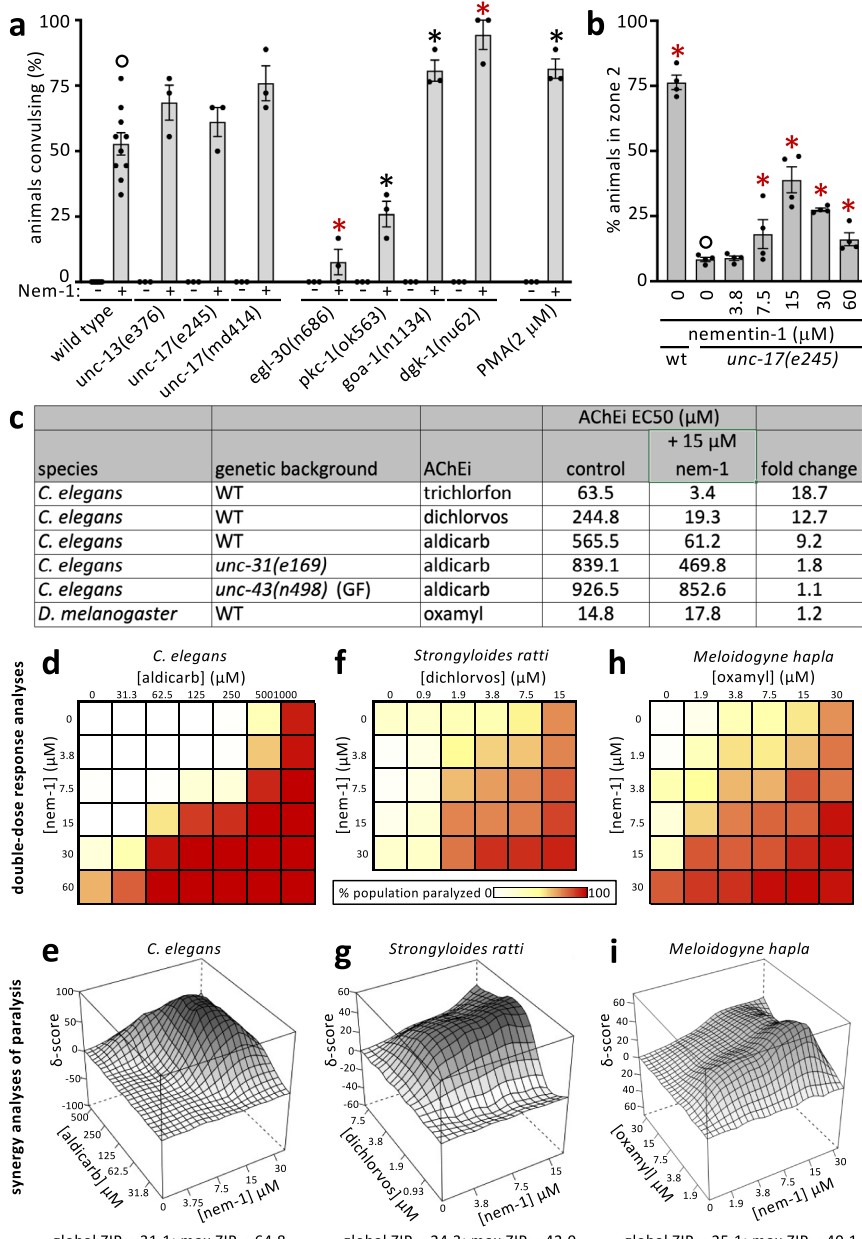

**Fig. 5 Nementin enhances AChE-inhibitor activity via DCV release. a** Convulsions induced after 80 min of exposure to nementin-1 (Nem-1) (60 μM) or vehicle control in the background of the indicated mutant gene, or in the background of 24 h of preincubation with the DAG memetic PMA. Showing the mean and SEM of $n = 3$ biologically independent trials ($n = 10$ for 60 μM wild-type) measuring 18 animals per trial. **b** The ability of wild type (WT) or *unc-17* mutant animals to productively locomote to zone 2 over 180 min after being placed in the center of a plate (see schematic in Fig. 3b) in the indicated nementin concentration. Showing the mean of $n = 4$ biologically independent trials with ~150 animals per trial. For both A and B, black asterisk $p < 0.01$, red asterisk $p < 0.001$, Bonferroni corrected Chi-square test of independence relative to control (open circle); SEM is shown. **c** Acetylcholinesterase inhibitor (AChEi) EC50s for different species and genetic backgrounds. Oxamyl is the only AChEi tested that exhibited activity against flies. EC50s are generated from a concentration series tested over $n = 3$ biologically independent trials scoring 18 animals per condition. **d**, **f**, and **h** Double dose-response analyses of nementin-1 (*y*-axis) vs the indicated AChEi (*x*-axis) in the indicated nematode species. The scale for the percentage of the population paralyzed applies to **d**–**f**. Each concentration was tested over $n = 3$ biologically independent trials for *elegans* with 18 animals within each trial, $n = 6$ trials for *ratti* with ≥30 animals within each trial, $n \geq 3$ trials for *hapla* with ≥15 animals within each trial. **e**, **g**, and **i** The corresponding zero-interaction potency (ZIP) synergy score heatmaps for the double dose-response analyses presented in **d**–**f**. Max ZIP-score is defined as the maximal ZIP-score obtained from a combination of three concentrations of each compound in the double dose response. Shading highlights the topology of the plot and differs between **g**–**i**. Primary data for this figure can be found in Supplementary Data 5.

## Cholinergic enhancement is a secondary effect of DCV release.

Neuropeptides released from DCVs are known to interact with G-protein coupled-receptors (GPCRs) that signal through G-proteins[32]. In *C. elegans* neurons, the Gα$_q$ protein EGL-30 stimulates DAG production via the EGL-8 and PLC-3 phospholipases[33] (Fig. 4c). By contrast, the Gα$_o$ protein GOA-1 reduces DAG abundance by triggering its phosphorylation by diacylglycerol kinase DGK-1[34]. Neuronal DAG primes both DCV and SV release via interaction with PKC-1(PKC-ε/η)[35] and UNC-13[36], respectively. The bioamines and neuropeptides

that are released from the DCVs can have both paracrine and autocrine activity[37,38]. This circuitry raised the possibility that nementin-1-induced DCV release may initiate a positive feedback loop that amplifies DCV release and enhances SV release as a secondary consequence (see green and yellow pathways in Fig. 4c).

We tested this hypothesis in three ways. First, we examined the impact of mutant components of the 'green' circuit depicted in Fig. 4c. We found that egl-30 RF, which reduces DAG production, suppresses nementin-1 convulsions (Fig. 5a). By contrast, goa-1 RF and dgk-1 loss-of-function (LF) mutants, which lack regulation of membrane DAG, enhance nementin-1 convulsions. We also found that disruption of PKC-1 reduces nementin-1 convulsions (Fig. 5a). Second, we asked whether the cell permeable DAG mimetic phorbol myristate acetate (PMA) enhances nementin-1 convulsions and found that it did (Fig. 5a). Consistent with the lack of convulsions in goa-1 RF mutants (not treated with nementin; Fig. 5a), PMA treatment (without nementin) does not induce convulsions in wild-type animals (Fig. 5a). This indicates that PKC-1-agonism alone cannot severely disrupt motor activity. Finally, we asked whether nementin-1's enhancement of cholinergic signaling is dependent on the machinery needed for DCV release. Indeed, we found that the unc-31(e169) RF mutation and the unc-43(n498) GF mutation suppress nementin-1's enhancement of cholinergic signaling (Fig. 5c). Together, these observations are consistent with a model whereby nementin-1 stimulates DCV release whose contents may stimulate autocrine/paracrine G-protein signaling, which in turn enhances both more DCV release and cholinergic signaling.

**Nementin disrupts parasitic nematodes of plants and animals**. We carried out a small structure-activity relationship (SAR) analysis of nementin analogs assayed against free-living nematodes, parasitic nematodes, and non-target models (Fig. 6; Supplementary Data 6). We purchased 18 commercially available analogs of nementin-1 and found that nementin-12 was the most potent inducer of acute motor phenotypes (Fig. 6). We expanded the SAR by synthesizing 22 analogs of nementin-12 (see Supplementary Methods). Nementins 1, 12-5, 13 and 14 demonstrated broad anthelmintic activity without inducing obvious phenotype against non-target organisms. In addition, nementin analogs were identified with improved activity in each parasitic nematode model tested. These data suggest that the nementin scaffold may be further refined to improve selectivity and potency.

Given that nementin-1 synergistically enhances the activity of AChE inhibitors in C. elegans, we tested whether it might act similarly against parasitic nematodes. We tested nementin-1's interaction with the organophosphate dichlorvos against Strongyloides ratti, a nematode parasite of mammals. Dichlorvos is approved for anthelmintic use in mammals[39,40]. Indeed, we find that nementin-1 synergistically enhances dichlorvos' activity against S. ratti (Fig. 5f, g). Nementin-1 was also found to synergistically paralyze the plant-parasitic nematode Meloidogyne hapla with the carbamate AChE-inhibitor oxamyl, which is approved for field use[39,40] (Fig. 5h, i). The nementins may therefore have added benefit against parasitic nematodes when used in combination with pesticides that are AChE inhibitors.

## Discussion
Here, we have developed a C. elegans-based screening approach to identify anthelmintic small molecules with phylum selectivity. Our approach was designed to yield broad-acting anthelmintics that acutely affect nematode behavior. The strategy yielded four small molecules that exhibit phylum selectivity within the limited range of organisms tested. We focused on one of these molecules, called nementin-1, which induces

C. elegans hyperactivity within seconds of exposure, followed by spastic convulsions minutes later. Several nementin analogs affect parasitic nematodes from distinct clades at concentrations that fail to elicit obvious phenotypes from non-target organisms. We were not able to generate C. elegans mutants that resist the lethal effects of nementin, suggesting that resistance in the wild may be difficult to achieve.

Our data support a model whereby nementin exposure initiates DCV release, which in turn enhances SV release that promotes cholinergic signaling. Nementin-induced convulsions depend on DCV release but not cholinergic signaling, indicating that DCV release is key to nementin's ability to disrupt motor behavior. How nementin stimulates DCV release remains unclear. Despite the phenotypic similarities between nementin exposure and unc-43 RF mutants, our data suggests that nementin is unlikely to inhibit UNC-43 in any canonical fashion. However, because of CaMKII's complexity and its many isoforms[14,41], we cannot rule out the possibility that nementin interacts with CaMKII in a complex and perhaps tissue-specific manner.

Convulsions in C. elegans and other animals are thought to manifest from an imbalance of excitatory and inhibitory neurotransmitter signals, termed E/I imbalance[42]. Gain-of-function mutations in motor neuron cation channels like the ACR-2 acetylcholine receptor or the UNC-2 voltage-gated calcium channel simultaneously increase excitatory cholinergic signals and decrease inhibitory GABAergic signals that are relayed to muscles. This in turn generates an excitation-dominated E/I imbalance that culminate in convulsions[18,19,38]. Several parallels exist between nementin-treated wild-type worms and either the unc-2 and/or acr-2 gain-of-function mutants including hyperactive locomotion, convulsions, a phenotypic dependence on UNC-31 CAPS, and aldicarb hypersensitivity[18,19,38]. Nementin-induced convulsions are independent of ACR-2, UNC-2, and robust cholinergic signaling. We therefore deduce that the likely E/I imbalance induced by nementin must originate from signals that are released from neuronal dense core vesicles.

Nementin has several features that make it an attractive anthelmintic for further development. First, in vitro tests demonstrate that it exhibits potentially broad-spectrum activity while maintaining phylogenetic selectivity. Such a feature is key to the development of an environmentally safe anthelmintic. Second, nementin analogs exhibit an in vitro potency that is comparable to several commercial anthelmintics. Third, the nementin synthesis route is relatively simple and uses inexpensive starting materials ('Chemistry' in Supplementary Materials)[43]. Finally, nementin has the potential to reduce the amount of non-selective pesticides that are notoriously released into the environment. The concentrations of nementin that are effective against C. elegans are likely to be different when used against parasites in the field[44,45]; further experimentation is required to test this hypothesis. In addition, it is currently unclear whether nementin can incapacitate drug-resistant parasitic strains, despite showing that drug-resistant C. elegans mutants remain sensitive to nementin (Supplementary Fig. 2; Supplementary Data 7).

Despite the toxicity concerns over indiscriminate activity, many AChE inhibitors such as trichlorfon, dichlorvos, coumaphos, aldicarb, fosthiazate, chlorpyrifos and oxamyl remain approved for nematicidal field use as of 2021 because of limited control options[39] (see the FDA Approved Animal Drug Products at https://tinyurl.com/by83ks98, the US Environmental Protection Agency at https://tinyurl.com/3n3kf34n; and the US Geological Survey pesticide study at https://tinyurl.com/bp5h5em4). Because of nementin's selectivity and synergistic effects, using it in combination with AChE inhibitors has the potential to maintain nematode-selectivity whilst simultaneously reducing the amount of non-selective pesticide applied to the environment.

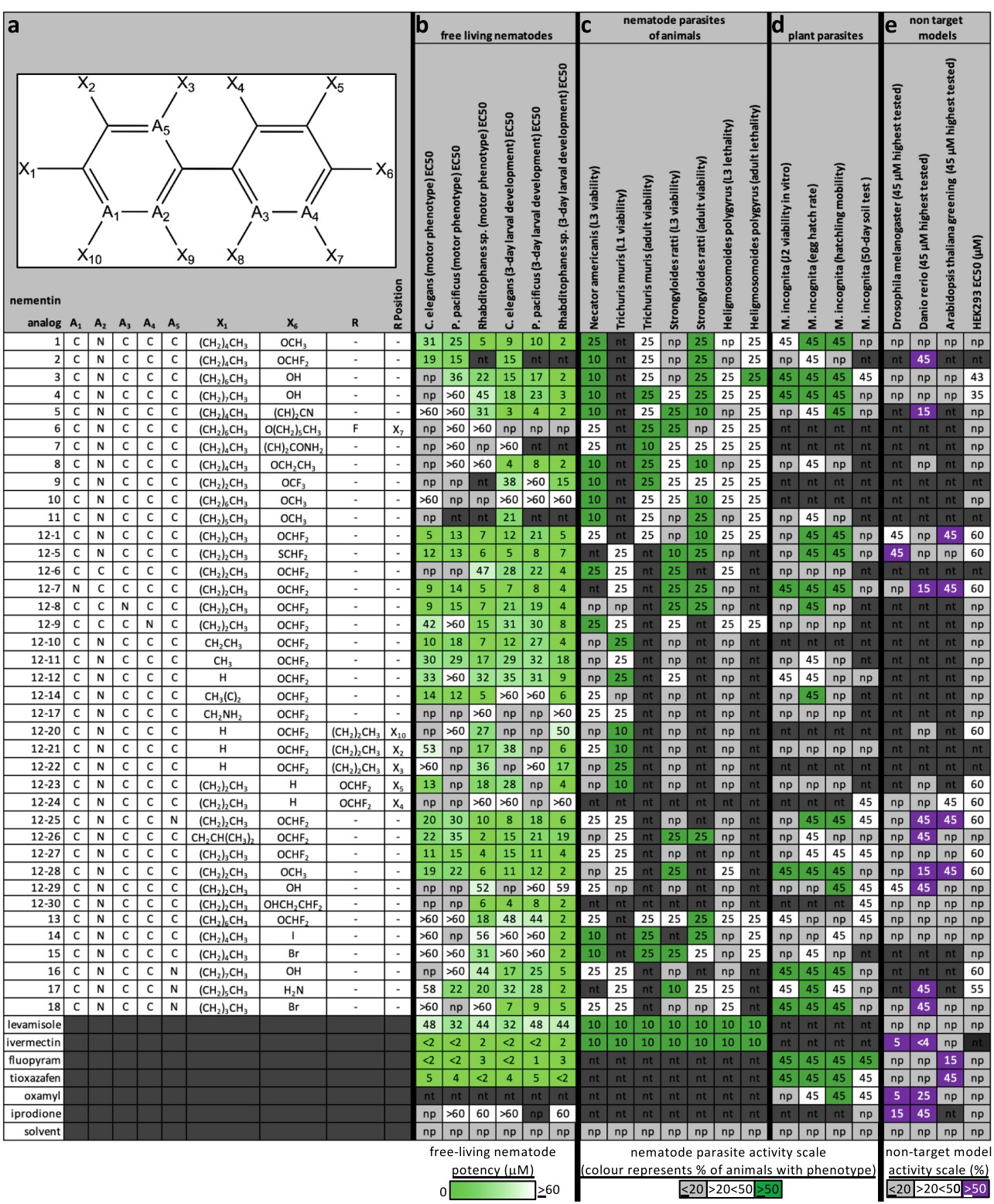

**Fig. 6 Nementin structure-activity relationship analysis. a** The analog structures are represented by the Markush structure at the top and includes atom and substituent identifiers. **b** Data for the free-living nematodes are EC50s (μM) and color-coded based on the potency scale at the bottom of this section. EC50s are generated from tests of a concentration series over $n = 3$ biologically independent trials. **c, d** Nematode parasites of mammals or plants were tested with a limited number of compound concentrations. In each cell in these sections, the lowest concentration (μM) tested that exhibited a phenotype is shown, and the percentage of animals that exhibit the phenotype is indicated by the color scale at the bottom of this section. **e** Data for models of non-targeted systems is reported either as an EC50 (μM) (for the HEK293 cells) or as the lowest concentration tested (μM) that exhibited a discernible phenotype compared to wild-type at the indicated concentration; the percentage of animals that exhibit the phenotype is indicated by the color scale at the bottom of this section. np no phenotype (<20% effect), nt not tested. See the Methods for a description of each assay. Primary data for this figure can be found in Supplementary Data 6.

## Methods

**Free-living nematode strains and culture**. All nematode strains were cultured using standard methods at 20 °C unless otherwise indicated[13]. The N2 (wild-type) strain of *Caenorhabditis elegans*, *Caenorhabditis briggsae* strain AF16 and *Pristionchus pacificus* strain PS312 were all obtained from the *C. elegans* Genetic Center (CGC; University of Minnesota). *Rhabditophanes sp. KR3021* was obtained from Marie-Anne Félix (Institute of Biology of the Ecole Normale Supérieure (IBENS), Paris, France). Mutant *C. elegans* strains were also obtained from the *C. elegans* Genetic Center.

**C. elegans small molecule screens and phenotypic analyses**. A library of 486 small-molecules (Chembridge) previously found to induce phenotypes in *C. elegans* (486 worm actives, aka wactives; 26108372) were tested for their ability to modulate *C. elegans* egg-laying. Drug dilutions were prepared from stock plates using a 96-well pinner tool (FP3S200 V&P Scientific, Inc.) transferring 0.3 mL of drug stock solution prepared in DMSO.

**C. elegans egg-laying assay**. To screen for small molecules that modulate egg laying, ~20 young-adult wild-type (N2 Bristol) animals were pipetted in 15 mL of M9 buffer to 96-well flatbottom polystyrene plates (2024-06 TC plate—Sarstedt) to a final volume of 50 mL with 60 μM of test molecules in either (1) M9 buffer to identify egg-laying stimulators; or (2) a combination of 12.3 mM serotonin creatine sulfate monohydrate (H775—Sigma-Aldrich) and 7.7 mM nicotine (N3876—Sigma-Aldrich) in M9 buffer that induces a robust egg-laying response to identify egg-laying inhibitors (aka NS condition). Screen molecules were transferred to test plates using a 96-well pinner tool (FP3S200 V&P Scientific, Inc.) transferring 0.3 mL of drug stock solution prepared in DMSO. Plates were incubated for 1 h at room-temperature. After 1 h 182 mL of solution containing 50 mM sodium azide (71289—Sigma-Aldrich) and 0.25% sodium dodecyl sulfate (SDS001.100 - BioShop) in M9 buffer using a multichannel pipette followed by 182 mL of M9 buffer to raise the volume of each well such that a flat meniscus is produced. Plates were immediately imaged on the *2020 Imager* and were separated by 5 min to allow time for preparation of subsequent plates for imaging. After imaging egg-laying data was extracted from raw captured images using the 'Egg & Worm Counter' ImageJ plugin described above. Stimulators were considered molecules that stimulated egg-laying ≥2-fold greater than two proximal controls in the benign M9 buffer conditions with significance ($p < 0.05$ unpaired heteroscedastic *t*-test) over triplicate measurement (3 test compared against 6 control wells). Egl inhibitors were considered molecules that suppressed egg-laying ≤ 0.5 the normalized egg-laying rate of proximal controls in the 'nicotine (7.7 mM) + serotonin (12.3 mM)' condition described above with significance ($p < 0.05$ unpaired heteroscedastic *t*-test) over triplicate measurement (3 test compared against 6 control wells). Primary egg-laying modulators that reached the above threshold in at least 1 of 2 additional tests were considered bonafide 'Egl modulators'. As a more stringent criteria to narrow our focus on robust Egl modulators we limited our inquiry of Egl stimulators that induced Egl ≥ 2 fold that of control over 3 trials (primary screen + 2 retests) or stimulated Egl 3-fold relative to control over at least 2 of 3 trials and Egl inhibitors that suppressed Egl ≤ 2 fold control over all 3 re-tests or molecules that suppressed Egl ≤ 3 fold that of control in at least 2 of 3 trials.

**Construction of the '2020 Imager'**. High-content brightfield data were acquired on a custom brightfield 96-well plate imager. The well-plate was mounted on a stationary platform, while the imaging setup traveled parallel to the bottom of the well-plate via a motorized stage. The plate was illuminated from above using a 10 W white LED. Underneath the plate, an Olympus 4x objective (Olympus, UPLFLN4XPH) was used with a 150 mm tube lens (Thorlabs, AC254-150-A-ML) to create an effective magnification of 3.33x. A mirror was used to maintain a low-profile imaging setup and minimize distortions induced by the displacement of the imaging optics. The magnified image was projected onto a 4 K line scan camera (Dalsa, P2-23-04K40), resulting in an effective resolution of 3.00 μm/px. The camera had a single line of 4096 px which captured an area of 12.3 mm by 3 μm with a bit depth of 10 bits. To capture a row of 12 wells, the image acquisition software (EPIX Inc, XCAP-Ltd) was setup to acquire scans at 3000 lines/s. This resulted in a 4 K by 48 K image that was acquired in about 15 s. To sequentially image each row in a well-plate, an Arduino UNO was used to synchronize the image acquisition with the movement of the motorized stage. For additional details see Aaron Au's Master's thesis titled: 'Optical Imaging Strategies for High-Content Studies of Development' available through the URL https://tspace.library.utoronto.ca/bitstream/1807/91538/3/Au_Aaron_K_201811_MAS_thesis.pdf.

**ImageJ analysis of C. elegans egg-laying rate**. A custom ImageJ (version 1.52i) plugin was used to quantify the number of worms and eggs present in each well (versions used available on github: github.com/seanph16/WormScanner3000/upload). Prior to object counting a threshold was applied based on the mean pixel intensity in each well image. The number of worms in a well was determined by measuring the area covered by non-egg shaped objects divided by the average area of a worm. Worm-like objects were recognized by the built-in ImageJ analyze particles function identifying objects >11450 px in area with a circularity of

0.029–0.80 (the approximate minimum single adult worm size and range of shape circularities adopted by a worm) divided by the median worm area. Single eggs were counted by first creating a mask of egg and egg clump shaped objects (objects that are 300–6000 px in area with circularity of 0.15–1.00), applying the built-in ImageJ 'Watershed' function to recognize single eggs within clumps and counting objects with a circularity of 0.5–1 with a size of 300–2500 px. The number of egg objects divided by the number of worms was used as a read-out of egg-laying behavior.

**C. elegans locomotory survey**. Locomotor phenotype analyses were done in 24-well plates with 1 mL of MYOB substrate (27.5 g Trizma HCl, 12 g Trizma Base, 230 g bacto tryptone, 10 g NaCl, 0.4 g cholesterol (95%)) seeded with 25 μL of OP50 *Escherichia coli* on each well. Each compound was added to the MYOB substrate before pouring to achieve the desired final concentrations of 30 μM or 60 μM after diffusion through the media. The final concentration of dimethyl sulfoxide (DMSO) in each of the wells was 1% v/v. Young adult or late fourth-staged larval worms are transferred into each well using a platinum wire pick. A Leica MZ75 stereomicroscope was used to visualize the movement of worms on the solid substrate. The specific locomotor phenotype (i.e. 'rubber-band' or 'coiler') was noted, and a qualitative assessment of the severity was made based on the degree of locomotor incapacitation and penetrance of the phenotype. Samples deemed 'Severe' indicated a strong perturbation and high penetrance, 'moderate' indicated a strong phenotype with low penetrance or a weak phenotype that is highly penetrant, and 'mild' indicated a weak phenotype that has low penetrance. Paralysis was distinguished from death by the presence of pharyngeal pumping.

*C. elegans Motor Phenotype Analyses:* The intensity of nementin-induced convulsions is tightly correlated with the degree of paralysis (i.e. animals that exhibit paralysis invariably convulse). We therefore used the degree of paralysis as a conveniently measurable proxy of convulsion intensity. In our survey of the effects of nementin analogs across *C. elegans*, *P. pacificus* and *R. diutinus*, animals were scored as convulsive if they failed to back at least ½ a body length after a touch on the head with a platinum wire. A more stringent scoring method was employed to compare convulsive phenotypes exhibited by *C. elegans* mutants. Animals were scored as convulsive if animals failed to demonstrate a sinusoidal wave form before or after a touch on the head with a platinum wire and failed to back ½ a body length. Like the convulsion scoring method above, animals treated with acetylcholinesterase inhibitors that failed to demonstrate a sinusoidal wave form before or after a touch on the head with a platinum wire that also fail to back ½ a body length were scored as paralyzed.

**Locomotory radiation assay**. ~150 L4/young-adult worms were pipetted in a 15 μL droplet onto the center of a standard 10 cm round culture plate containing MYOB media + agar containing small-molecule with 1% DMSO. Media with drug were prepared in 50 mL falcon tube inverted 10x before pouring. Plates were dried for 90 min before being supplemented with a full lawn of OP50 bacteria seeded from a saturated culture of OP50 grown in LB broth. Plates were left uncovered adjacent to a flame until the (~15 min). Three hours after pipetting worms onto plates the plates were flash frozen at -80 °C for 3 min to freeze worms in place. The fraction of worms that traveled at least 1.65 cm (diameter after measuring 1 cm from the edge of the droplet) from center of the plate were recorded.

**Developmental growth assay**. *C. elegans* larval development assays were conducted in 96-well flatbottom clear flatbottom plates. ~20 L1 larvae in 10 μL of M9 buffer were pipetted into each test wells containing 40 μL of NGM media supplemented with HB101 *E. coli* with the desired test compound (+0.6% dimethyl-sulfoxide (DMSO; Sigma-Aldrich product ID: D8418) as the chemical solvent). Plates were wrapped in 3 layers of brown paper towels soaked with water. After either 3 or 6 days of incubation the number of *C. elegans* animals of different larval stages were recorded using a Leica MZ75 stereomicroscope.

**C. elegans confocal microscopy**. *C. elegans* KG4247 expressing ceIs201 [unc-17p::ins-22::Venus + unc-17p::RFP + unc-17p::ssmCherry + myo-2p::RFP] were incubated on 6 cm MYOB media + 2% agar plates containing either 60 μM Nementin-1 or 60 μM Nementin-12 with 1% DMSO or 1% DMSO alone for 4 h at room temperature. Wells were seeded with OP50 *E. coli* bacteria and used the day after preparation (see locomotory survey for further details on the preparation of media + drug plates). After incubation, animals were picked onto a 5% agar pad, 10 μL 10 mM tetramisole hydrochloride (prepared from 99% (−)-tetramisole hydrochloride, Sigma-Aldrich product ID: L9756) solvated in standard M9 buffer was pipetted onto the pad and a cover glass put on top. Animals were imaged using a Leica DMI 6000 B confocal microscope with a Hamatsu C9100-31 camera with a 100x oil immersion objective. A 491 nm laser was used to excite INS-22::Venus and images were captured with 25 ms of exposure. A 510 nm laser was used to excite ssmCherry and RFP and images were captured with 100 ms of exposure. Images were captured after anterior and dorsal nerve cord features were brought into focus in the red channel (the RFP remained stable for the duration of imaging). Anterior, midbody and posterior regions containing respective coelomocytes (ccPR + ccAR, ccPL + ccAL & ccDL respectively) were captured. Images were captured over a 30 μm Z-stack captured with a 0.5 μm step and all images were captured within

25 min of slide preparation. A maximal projection containing the ventral and dorsal nerve cords and coelomocyte was generated for each captured section in ImageJ (version 1.52i). Tracings of captured axonal sections and coelomocytes were manually drawn and fluorescence signal measured in ImageJ. Due to the variability in coelomocyte endocytic/lysosomal vesicle content, coelomocytes were reported as the measurement of mean fluorescence signal of the GFP channel compared to the RFP channel. For axonal segments, the mean of two measurements of each region and representative background were collected to adjust for variability in manual measurement. Regions of interest for at least 15 animals were captured over several imaging sessions, at least 3 control animals were captured in each imaging session.

## Parasitic nematode assays

*Cooperia oncophora assay.* Fresh cattle feces containing eggs of an ivermectin-resistant strain of *C. oncophora* were kindly supplied by Dr. Doug Colwell and Dawn Gray (Lethbridge Research Station, Agriculture and Agri-Food Canada). Established methods were used to carry out the experimental cattle infections, and these methods were approved by the Lethbridge AAFC Animal Care committee and conducted under animal use license ACC1407. Cattle feces containing *C. oncophora* eggs were stored anaerobically at room temperature for a maximum of 6 days before use. Eggs were isolated from feces using a standard saturated salt flotation method immediately before the egg hatch assay. 80 μl of distilled and deionized water was added to each well of a 96-well culture plate, and then 1 μL of chemical at the appropriate concentration in DMSO was added to each well using a multichannel pipette. Approximately 50 eggs were added per well in 20 μL of water for a final volume of 100 μL in each well; the final DMSO concentration was 1% (v/v). The eggs were incubated in the chemicals for 2 days at room temperature, after which hatching was stopped by the addition of 1 μL iodine tincture to each well. The number of hatched larvae was counted at each concentration, and eggs that failed to hatch were scored as dead. *Relative viability* values were calculated by dividing the fraction of eggs that hatched at each concentration by the fraction of eggs that hatched in the corresponding DMSO control well. Two biological replicates were performed for each dose-response experiment, and the relative viability values were averaged across the biological replicates. The average hatch rate for the DMSO control wells was >93% for both biological replicates.

*Dirofilaria immitis assay.* Experiments on *D. immitis* microfilariae were performed in the laboratories of Bayer Animal Health GmbH (Monheim, Germany). The Missouri *D. immitis* isolate used for all assays was originally isolated from an infected dog from Missouri (USA). From 2005 onwards, the isolate was maintained and passaged in beagle dogs at the University of Georgia (Athens, GA, USA). From 2012 onward, the isolate was also maintained at the laboratories of Bayer Animal Health GmbH in Monheim, Germany. For the experiments with microfilariae, blood was sampled from beagle dogs (Marshall BioResources, North Rose, NY, USA) with patent infections, and microfilariae were purified according to the protocol described by the FR3.

Approximately 250 freshly purified microfilariae were cultured in single wells of a 96-well microtiter plate containing supplemented RPMI 1640 medium. Microfilariae exposed to medium substituted with 1% DMSO were used as negative controls. Motility of microfilariae was evaluated after 72 h of drug exposure using an image-based approach—DiroImager, developed by Bayer Technology Services. This device is a fully automated high-throughput platform, allowing high-resolution optical imaging of an entire 96-well microtiter plate. The DiroImager integrates a high-resolution video camera (Prosilica GT6600; Allied Vision) with a telecentric lens (S5LPJ3005; Sill Optics) that prevents perspective distortion of the recorded images, ensuring high accuracy of measured values across all samples. In brief, a series of 20 high-resolution images were recorded (one per second). In a first step, image processing filters were used that discriminate larger objects to avoid the detection of crystallized or undissolved particles. In the actual calculation, pixel-wise differences between sequential images were calculated to determine worm movement between single images of a series; test compound activity was determined as the reduction of motility in comparison to the solvent control. Based on the evaluation of a wide concentration range, concentration–response curves as well as IC50 values were calculated were applicable.

*Nippostrongylus brasiliensis acetylcholine esterase secretion assay.* This assay has been previously described in detail[46]. AChE is secreted by many parasitic nematodes, including *N. brasiliensis*. Assaying a small molecule's impact on AChE secretion is therefore a proxy for its ability to modulate the nematode's nervous system. Methods have been previously developed to assay AChE secretion from *Nippostrongylus* using colourimetric determination of AChE activity in the culture medium[47]. Briefly, test compounds were dissolved in DMSO at a concentration of X, Y, Z and serial dilutions were performed in DMSO resulting in stock solutions of A, B, C. Stock solutions were stored at −20 °C until they were diluted 1:200 with culture medium (20 g/l Bacto Casitone, 10 g/l yeast extract, 5 g/l glucose, 0.8 g/l KH2PO4, 0.8 g/l K2HPO4, 10 μg/ml sisomycin and 1 μg/ml clotrimazole, pH 7.2). Final drug concentrations were E, F, G μM in 1.0% DMSO. Because secretion of AChE is gender and body weight specific two female and three male adult worms were placed in each well containing 1 ml of pre-warmed medium with drugs plus vehicle and incubated at 37 °C and 95% relative humidity for 5 days[48]. All drug concentrations were performed in duplicate. From each well 25 μl medium were

transferred into a 96-well plate. Then, 250 μl 5,5′-dithio-bis (2-nitrobenzoic acid) (0.25 μM) and 25 μl acetylthiocholine (4 mM) were added. AChE cleaves acetylthiocholine into acetate and thiocholine. In a consecutive reaction, thiocholine reacts with 5,5′-dithio-bis(2- nitrobenzoic acid) to thionitrobenzoate. Thionitrobenzoate is a yellow dye and its concentration can be determined by measuring the absorption at 405 nm. The A405 was measured after two and 7 min of incubation at RT using an Expert 96 plate reader (Asys-Hitech, Salzburg, Austria) and the software MikroWin 2000 (Mikrotek, Overrath, Germany). The difference in absorption between both time points was taken as measure of AChE activity. The arithmetic mean of 12 no drug control wells was set to 100% activity, and reduction of AChE activity in percentage relative to the negative control was calculated for each test compound concentrations. Within an assay, every drug concentration was performed in duplicate, and the software reported the mean of these duplicates.

*Strongyloides ratti L3 larvae lethality.* Data are the measurement of the % of Larval stage 3 (L3) worms (as indicated) that respond to 80 °C hot water stimulus after 24 h or 72 h of incubation in wells containing the indicated compound. Data are the mean measurement of 30-40 larvae incubated in a dark box at room temperature for 24 or 72 h over duplicate biological replicate conducted in triplicate.

*Trichuris muris L1 larvae experiments.* *T. muris* eggs were collected from the feces of the infected mice (as described above) using a flotation method with saturated NaCl solution in Milli-Q water. *T. muris* eggs were stored in Milli-Q water in the dark for 3 months at 23–25 °C, until the eggs were embryonated. *T. muris* L1 were obtained using a hatching procedure with *E. coli*[49]. 30-40 larvae were placed in each well of a 96-well plate containing 175 μl culture medium and 25 μl of the test drug stock solutions. Larvae were kept at 37 °C, 5% CO2 for 24 h. To evaluate the drug effect first the total number of L1 per well was determined. Then, 50-80 μl of hot water (≈80 °C) was added to each well and the larvae that responded to this stimulus were counted. The proportion of larval death was determined. Larval survival counts were averaged over duplicate biological replicate conducted in triplicate normalized to controls.

*T. muris adult experiments.* Mice (C57BL/6NRj) were infected with 200 embryonated *T. muris* eggs. Seven weeks post-infection *T. muris* adult worms were collected from the intestines. Three worms were placed in each well of a 24-well plate containing 1980 μl culture medium and 20 μl of the test drugs (10 μM of a 1 mM stock solution). After 72 h of incubation at 37 °C, 5% CO2 the condition of the worms was microscopically evaluated using a viability scale from 3 (normal activity) to 0 (dead). Viability scores were averaged across replicates and normalized to the control wells. The experiment was conducted in duplicate.

*Heligmosomoides polygyrus L3 viability.* *H. polygyrus* infection 3-week-old female NMRI mice were obtained from Charles River (Sulzfeld, Germany). Rodents were kept under environmentally controlled conditions (temperature: 25 °C, humidity: 70%, light/dark cycle 12 h /12 h) and had free access to water (municipal tap water supply) and rodent food and were allowed to acclimatize for 1 week. NMRI mice were infected with 88 *H. polygyrus* L3. Two weeks post-infection, mice were dissected cultivating the eggs on an agar plate for 8–10 days in the dark at 24 °C. For the assays, 30–40 larvae were placed in each well of a 96-well plate containing 175 μl culture medium and 25 μl of the test drug stock solutions. *H. polygyrus* adults and stage 3 larvae (L3) were incubated in RPMI 1640 (Gibco, Waltham MA, USA) medium supplemented with 5% amphotericin B (250 μg/ml, Sigma-Aldrich, Buchs, Switzerland) and 1% penicillin 10,000 U/ml, and streptomycin 10 mg/ml solution (Sigma-Aldrich, Buchs, Switzerland). Culture plates were kept in a dark box at room temperature for up to 72 h. To evaluate the drug effect first the total number of L3 per well was determined. Then, 50-80 μl of hot water (≈80 °C) was added to each well and the larvae that responded to this stimulus were counted. The proportion of larval death was determined. Larval survival counts were averaged over duplicate biological replicate conducted in triplicate normalized to controls.

*Necator americanus L3 viability.* *N. americanus* larvae (L3) were obtained by filtering the feces of infected hamsters and cultivating the eggs on an agar plate for 8–10 days in the dark at 24 °C. *Necator americanus* L3 were incubated in Hanks' balanced salt solution (HBSS; Gibco, Waltham MA, USA) supplemented with 10% amphotericin B and 1% penicillin (10,000 U/ml) and streptomycin (10 mg/ml) solution. For the assays, 30–40 larvae were placed in each well of a 96-well plate containing 175 μl culture medium and 25 μl of the test drug stock solutions. Treated Larvae were kept in a dark box at room temperature for up to 72 h. To evaluate the drug effect first the total number L3 per well was determined. Then, 50–80 μl of hot water (≈80 °C) was added to each well and the larvae that responded to this stimulus were counted. The proportion of larval death was determined. Larval survival counts were averaged over duplicate biological replicate conducted in triplicate normalized to controls.

*Meloidogyne incognita assays.* *M. incognita* infective second stage juvenile (J2) in vitro viability assays were performed in 96-well polystyrene plates. Each well contained ~25 J2s and compounds were added at a final concentration of 45 μM (0.5% DMSO v/v) in a total volume of 100 μL of sterile distilled water. Plates were

sealed with parafilm and incubated for 72 h at 25 °C. At the end point the fraction of viable nematodes in each drug condition and DMSO solvent controls was calculated by dividing the number of mobile nematodes by the total number of nematodes in the well. The experiment was conducted twice, with three technical replicates per treatment in each trial. *M. incognita* egg hatching assays were performed in sterile distilled water in 96-well plates similarly to the J2 viability assays described. Embryos were incubated in 45 µM (0.5% DMSO v/v) compound for 7 days at 25 °C. At the end point the number of hatched embryos was quantified in each condition and DMSO solvent controls. The fraction of hatched juveniles that were mobile was also quantified ('hatchling mobility'). The experiment was conducted twice, once with 50 embryos plated per well and once with 100 embryos plated per well, with three technical replicates per treatment in each trial. *M. incognita* 50-day soil reproduction assays were conducted in 90 g of soil (1:1 sand:loam mix) per compartment in 6-pack planting containers. The soil was drenched with 18 mL of deionized water containing dissolved chemical or DMSO solvent alone. Approximately 1500 J2s were inoculated into the soil in 2 mL of water, for a total volume of 20 mL. The J2s were incubated in the soil and chemical for 24 h after which a 2–3 week old tomato seedling was transplanted into the soil. Tomatoes were grown for 8 weeks in a greenhouse under long-day conditions (16 h photoperiod) with 26/18 °C day/night temperatures. At the end point of the assay the tomato roots were harvested and eggs were extracted by rinsing in 0.6% sodium hypochlorite solution with agitation at 300 rpm for 3 min. Roots were rinsed with water over nested sieves and eggs present in each root system were collected and quantified. Roots were dried in a 65 °C oven and the number of eggs per milligram of dried root material was calculated. The experiment was conducted twice, with two technical replicates per treatment in each trial.

*Meloidogyne hapla motor assay.* *M. hapla* motor assays were conducted using J2 infective larvae isolated from ornamental tomato plant roots. J2s were isolated by isolating egg masses from the root network of infected plants and hatching in deionized water at room temperature for ~1 week. 10 µL of deionized water containing ~15 J2s (no fewer than 10 J2s) were pipetted into 96-well polystyrene plates containing the drug condition of interest with 0.6% DMSO. Addition of J2s to wells were staggered by 35 s for the purpose of maintaining a stringent endpoint. Drug dilutions were prepared from stock plates using a 96-well pinner tool (FP3S200 V&P Scientific, Inc.) transferring 0.3 mL of drug stock solution prepared in DMSO. Animals were incubated at room temperature with shaking for 4 h (100 RPM; helps concentrate J2s in the middle of wells). At the 4 h endpoint, 30 s videos of each well were captured using a Leica FLEXACAM C1 USB camera mounted to a Leica MZ75 stereomicroscope using Leica LAS EZ image capture software (V3.4.0). Videos were sped up 5x and the number of body bends generated over 30 s was recorded; animals were scored as paralyzed if they failed to generate >1 body bend over the 30 s recording. Data is reported as the mean % of paralyzed J2s over 3 or 4 independent biological replicates with ~15 animals per well (wells containing <10 animals were not scored). The dose-response matrix reporting the mean % of paralyzed J2s was used as the input for the SynergyFinder 2.0 server (https://synergyfinder.fimm.fi/).

**Small-molecule tanimoto coefficient pairwise similarity.** Pairwise similarity scores were calculated as the Tanimoto coefficient of shared FP2 fingerprints using OpenBabel (http://openbabel.org). A Further description of Tanimoto pairwise similarity is provided in Burns et al. 2015 (26). Network visualization for Fig. 1e was performed using Cytoscape (version 3.7.2).

**Zebrafish chemical treatments and phenotypic analyses.** All phenotypic analysis was performed on a stereomicroscope. At 1 dpf, 5 embryos were placed in 1 mL filter-sterilized egg water with chemicals in sterile 24-well plates (Falcon). At 3dpf, larvae were anaesthetized with ~0.6 mM tricaine methanesulfonate (tricaine), mounted in 3% methylcellulose on glass slides and bright field images were taken with a 4x objective using a light microscope (Olympus BX43). The morphology of embryos relative to vehicle controls was assessed including their size, presence of edema, heart rate (normal, slow, or nearly absent), and presence of necrosis.

All chemicals were prepared in DMSO and added to filter-sterilized egg water at 0.1% of the final volume. Equal volumes of vehicle solvent were used in all conditions for a single assay. Note that methylene blue was not added to the egg water in any chemical assays. Culture plates were sealed with parafilm, wrapped in aluminum foil, and incubated at 28.5 °C until the assay date.

A photoactivation assay was used to elicit movement and assess locomotion of zebrafish larvae as previously described[50]. At 1 dpf, embryos in their chorions were aliquoted into 150 µL system water in 96-well plates (Falcon). Next, 50 µL of 4X chemical was added to each well to bring the volume to 200 µL and 1X final concentration (either 3.75–60 µM). Plates were incubated until 3dpf, at which time any embryos still in their chorions were manually dechorionated in their wells. To assay locomotion, 10 µL of 210 µM optovin analog 6b8 (ChemDiv ID#2149-0111 or ChemBridge ID#5707191) was added to each well for a final concentration of 10 µM, incubated for 5 min, and movement tracked on the ZebraBox platform (ViewPoint) using a 30 s lights on/off for 3m30s.

**Arabidopsis thaliana greening assay.** Greening experiments were performed with *Arabidopsis thaliana* seeds of wild-type Col-0; seeds were surface sterilized in

bleach and plated onto 0.5X MS, 0.5% sucrose agar medium supplemented with compounds of interest at 5, 15 and 45 µM concentrations (0.2% DMSO (v/v)). After 4d of stratification at 4 °C, plates were transferred to a growth chamber (16 h / 8 h, 150 µE/m$^2$) and greening recorded after 4 days. Pictures were recorded by camera (SONY a7s) with FE1.8/55 lens (FE 55 mm F1.8 ZA; SEL55F18Z). Experiments were performed in triplicate for each treatment.

**Drosophila melanogaster dose-response assay.** Fly food in agar substrate was prepared by mixing 100 mL of unsulfured molasses, 100 mL of cornmeal, 41.2 g of Baker's yeast, and 14.8 g of agar into 1400 mL of distilled deionized water and boiling for 30 min. The media was allowed to cool to 56 °C, at which point 5 mL was added by syringe to plastic cylindrical fly vials. 10 µL of chemical, or DMSO alone, was added to the media in each vial. The chemicals were mixed into the media by mechanical mixing using a pipette. The final DMSO concentration was 0.2% (v/v). The media was allowed to solidify at room temperature (~22 °C) overnight. The following day (Day 0), eight pairs of male and female w1118 flies were added to each vial so that there were 16 flies in total per vial. The vials were stored at room temperature for 7 days, at which point the number of mobile flies was counted. Fly mobility was scored as any observable movement after the vial had been vigorously jostled. *Relative mobility* was calculated by dividing the number of mobile flies in the treatment vials by the average number of mobile flies in two DMSO control vials. On Day 8 the 16 parental flies were removed from the vials and the progeny larvae were allowed to continue to grow and hatch into adult flies. To assess larval viability, hatched flies were counted and discarded on Days 10, 12, 14, 16, 18, and 20. The counts were summed. *Relative viability* was calculated by dividing the number of hatched flies in the treatment vials by the average number of hatched flies in the two DMSO control vials. The final relative mobility and relative viability values are an average across three experimental replicates.

**HEK293 proliferation assay.** HEK293 cells (Invitrogen's Flp-In-293 Cell Lines (catalog #R75007)) were seeded into 96-well plates, at 5000 cells per well, in 100 µL total volumes of DMEM/10%FBS/1%PS media and grown overnight at 37 °C in the presence of 5% CO$_2$. Compounds (0.5 µL volumes from appropriate source plates) were then added to cells, and growth was continued for an additional 48 h. Following growth, 10 µL of CellTiter-Blue Viability reagent (Promega) was added to each well, and plates were incubated for an additional 4 h at 37 °C in the presence of 5% CO$_2$. Fluorescence measurements (560 nm excitation/590 nm emission) were then performed using a CLARIOstar Plate Reader (BMG Labtech) to quantify reagent reduction and estimate cell viability.

**Spicule protraction assay.** L4 males grown overnight on Nematode Growth Media (NGM) 2% agar plates + OP50. Next day adult males plated in liquid NGM liquid culture with 1% DMSO or 60 µM Nementin-1 with 1% DMSO in polystyrene 96-well plates. Adults were observed under a Leica MZ75 stereomicroscope at indicated time points. Reported '% Protracted' includes partial spicule protraction.

**Statistics and reproducibility.** Unpaired one or two-sided *t*-tests or one-sided ANOVA with Dunnett's adjustment for multiple comparisons were conducted between control and treatment groups with where appropriate. Two-sided Chi-square tests with Bonferroni correction were conducted for comparison of proportional convulsion data to respective controls. Extra sum-of-squares F tests were conducted comparing EC50 curves generated for dose-response data. Statistical analyses were conducted using GraphPad Prism (version 9). Zero-Interaction Potency (ZIP) synergy scores and heatmaps were generated using the SynergyFinder2.0 server using the default parameter set.

**Reporting summary.** Further information on research design is available in the Nature Research Reporting Summary linked to this article.

## Data availability

Original data for all analyses presented are included in Supplementary Data 1–7. Original images for analyses can be made available per request through the corresponding author P.J.R.

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

## Acknowledgements

We are grateful for the strains given to us by the *Caenorhabditis* Genetics Center (University of Minnesota), Marie-Anne Félix (IBENS, Paris), Ken Miller (Oklahoma Medical Research Foundation), and Mei Zhen and Wesley Hung (Lunenfeld-Tanenbaum Research Institute, Toronto). Many thanks to Dr. Doug Colwell and Dawn Gray at the Lethbridge and Agri-Food Canada Research Centre for supplying feces from infected calves for *Cooperia oncophora* egg collection. Research in the I.Z. lab is partially supported by the United States Department of Agriculture, Agricultural Research Service. P.J.R. is supported by CIHR project grants (313296 and 173448). P.J.R. is a Canada Research Chair (Tier 1) in Chemical Genetics.

## Author contributions

Conceptualization, P.J.R., S.H.; Methodology, S.H., J.J.K., K.-L.C., A.A., M.K., C.H., J.P., J.R.V., M.G., J.S., V.W., B.M.P., E.M.R., A.S.V.; Investigation, S.H., J.J.K., A.B., M.K., C.H., J.P., C.D'.A., Y.-H.K., J.R.V., M.G., J.S., V.W., B.M.P., E.M.R., A.S.V.; Visualization, P.J.R., S.H.; Funding acquisition, P.J.R., I.Z.; Project administration, P.J.R.; Supervision, P.J.R., J.S.G., I.S., S.R.C., J.J.D., C.M.Y., J.K., I.Z., M.L.; Writing original draft, P.J.R., S.H., J.J.K., K.-L.C., A.A., S.R.C., D.K., J.K.; Draft Review & Editing, P.J.R., S.H., A.R.B., J.J.K., S.R.C., D.K., J.K., I.Z., M.L.

## Competing interests

The authors declare the following competing interests: S.H., J.K., A.R.B., K-L.C., J.P., M.L. and P.J.R. have patents pending related to nementin; the remaining authors declare no competing interests. Mention of trade names or commercial products in this publication is solely for the purpose of providing specific information and does not imply recommendation or endorsement by the U.S. Department of Agriculture.

## Ethics

DK: Experiments on *D. immitis* and *N. brasiliensis* were performed in the laboratories of Bayer Animal Health GmbH (Monheim, Germany) in accordance with the local Animal Care and Use Committee and governmental authorities. JK: The generation of *N. americanis*, *T. muris*, S. ratti, and *H. polygyrus* and in vitro studies were carried out at the Swiss Tropical Institute (Basel, Switzerland), in accordance with both cantonal (license no. 2070) and Swiss national regulations on animal experimentation. JD: All zebrafish experiments were performed in compliance with any relevant ethical regulations, specifically following an institutionally reviewed and approved animal use protocol as well as the policies and guidelines of the Canadian Council on Animal Care and Animals for Research Act of Ontario. USDA is an equal opportunity provider and employer.
