## [Peer Review File · Communications Biology]

Reviewers' comments:

Reviewer #1 (Remarks to the Author):

This is another important paper from a group that has made notable advances in the use of *C. elegans* as a platform for the discovery of novel anthelmintics and nematocides. It is generally well-written, with clear and concise descriptions of the assay and the data generated. It warrants publication in this journal after some minor revisions.

1. This is a continuation of work done to identify 'worm active' (wact) compounds in a high-throughput *C. elegans* screen, and I think the authors need to be clearer about this. The current assays are not so much a pipeline as an assay modification to identify valuable wact compounds identified in the previous work (as far as I can tell). Indeed, I think they should play up their previous successes in this regard.

2. Language use seems a bit odd in places. Examples: nementin is not a proper noun and does not deserve to be capitalized; Line 94: 'destroys' is incorrect; nematode parasitism lessens productivity of livestock, but is rarely lethal; Line 112 and elsewhere; agonism is a pharmacological concept referring to receptor activation, and 'agonizes' is not normally used in this context; 'stimulates' may be preferable; Line 117: nementin represents a potentially valuable scaffold (which is the alkyl phenylpiperidine moiety); Line 126: 'paragon' does not seem to be the right word here; Line 181: 'popular' should be replaced with 'commercially available'; not finding resistant mutants in *C. elegans* screens is a good thing but is not really proof that it will be slow to develop in the field (as I know the authors are aware; just need to be a bit more conservative in this case).

The discussion is reasoned and appropriately cautious, but lays out realistic scenarios around mechanism of action. The paper, as mentioned above, is important, and I congratulate the authors on their work.

Reviewer #2 (Remarks to the Author):

The presented work on the discovery approach based on behavior of nematodes and the presentation of some molecules with anthelmintic action is a very important contribution and needs to be published to become available to a broader community.

In particular, the approach on nematode behavior (motoric based) has been elegantly and convincingly laid out.

Some specific comments:

in significance statement:

the statement's 2nd sentence is too absolute - not all existing anthelmintics are being "rendered ineffective because of the evolution of resistance". This depends largely on the hosts (more a problem in livestock less in companion animals) and on the nematode species. I recommend the addition of "many" at the start of this sentence, which still would highlight the apparent urgency.

Results and discussion:

Important results are hidden and difficult to read:

analysis of Nementin analogs assayed against free-living nematodes, parasitic nematodes, and non-target models (Supplementary Table 3; Supplementary Data 4) - considering the statement that nementin is a nematode selective agonist and considering the significance statements that novel anthelmintics are needed for animal health and crop production, the results on real parasitic nematodes (and not only on *C. elegans*) must be visible in the manuscript, also in an easier to read summary table.

Interference with motoric action of nematodes is shown for the egg laying parameter. For animal pathogenic nematodes, motoric interference may be even more important for movement, due to necessity of nematodes to counteract against host gut peristaltic. The required concentrations may be different as is the case of macrocyclic lactones.

For application against plant pathogenic nematodes it might be of value to add a beneficial - non pathogenic soil nematode to confirm the selective activity.

To validate and possibly even more substantiate the value of the presented approach, the effect of nementin on ML- and levamisol-resistant nematodes could be presented.

Details:

pipeline or approach? pipeline (eg in introduction) in parasiticide discovery is defined as a summary of available molecules at various stages in the discovery process. In here the authors present a very valuable specific approach for discovery and on top present various molecules but focus on one, so "approach" maybe more applicable.

name -nementin:

is the name applicable to the WHO process of naming novel anthelmintics? If the class of molecules would become valuable for application, naming might be different.

I would recommend to consider some of my comments (for the authors) for some minor revisions, particularly adding results on true parasitic nematodes in an accessible and readable form.

If available, activity on drug resistant nematodes should be pointed out, if not, those needs to be discussed.

Overall, the MS should definitely be published with some minor revisions.

Reviewer #3 (Remarks to the Author):

The increasing global drug resistant problem in parasitic nematodes which cause huge human and animal diseases and enormous losses of crop production makes the new drug discovery and vaccine development an urgent task. This manuscript identified a compound named nementin by using a pipeline exploiting multiple motor outputs of the model nematode *C. elegans*. It is found that nementin-1 can induce worm's convulsions and paralysis by agonizing neuronal dense core vesicle release and cholinergic signaling. By consequence, nementin synergistically enhances the activity of nematode's AChEIs, implicating the potential development of nementin as an environment-friendly nematicide. The aim of this work is clearly formulated, the methods are presented in detail and the results are clearly described. I would suggest to accept for publication after a minor revision.

Line 218, delete "()".

Lines 225, "4" should be changed into "3".

Line 275, delete "." After "a". "UNC-1" should be read as "UNC-13".

Line 620, "(d-f)" should be changed into "(d, f, h)".

Line 622, "(g-i)" should be changed into "(e, g, i)".

Reviewers' comments:

Reviewer #1-General Comments: This is another important paper from a group that has made notable advances in the use of *C. elegans* as a platform for the discovery of novel anthelmintics and nematicides. It is generally well-written, with clear and concise descriptions of the assay and the data generated. It warrants publication in this journal after some minor revisions. The discussion is reasoned and appropriately cautious, but lays out realistic scenarios around mechanism of action. The paper, as mentioned above, is important, and I congratulate the authors on their work.

Our Response: We thank the reviewer for their kind words.

Reviewer #1- Comment 1: This is a continuation of work done to identify 'worm active' (wact) compounds in a high-throughput *C. elegans* screen, and I think the authors need to be clearer about this. The current assays are not so much a pipeline as an assay modification to identify valuable wact compounds identified in the previous work (as far as I can tell). Indeed, I think they should play up their previous successes in this regard.

Our Response: We thank the reviewer for the good suggestion. We have added the following paragraph to the revised introduction, followed by the modification of a few subsequent sentences:

In our efforts to identify novel scaffolds with potential nematicidal/anthelmintic utility, we previously screened over 67,000 drug-like compounds for those that disrupt the life cycle of the free-living nematode *Caenorhabditis elegans*⁷. We called the collection of 627 hits the worm-active (aka wactive) library. The wactive hits were re-screened against two parasitic nematodes and two vertebrate models, yielding 67 molecules belonging to 30 distinct small molecule scaffolds that selectively kill *C. elegans* and the parasitic nematodes *Cooperia oncophora* and *Haemonchus contortus*⁷. We showed that one of these scaffolds selectively kills nematodes via the inhibition of complex II (succinate dehydrogenase)⁷.

Here, we present a novel motor-centric screening pipeline to identify novel candidate nematicides. Disrupting a parasite's motor control has repeatedly proven effective in mitigating nematode infection^{8,9}. We therefore re-screened our wactive library using two successive behavioural assays of the free-living nematode *Caenorhabditis elegans*.

Reviewer #1- Comment 2: Language use seems a bit odd in places. Examples-nementin is not a proper noun and does not deserve to be capitalized

Our Response: We have changed nementin's capitalization throughout.

Reviewer #1- Comment 3: Line 94: 'destroys' is incorrect-nematode parasitism lessens productivity of livestock, but is rarely lethal;

Our Response: We have changed the line from, 'but nematode parasitism also destroys tens of billions of dollars (USD) worth of livestock annually¹.', to, 'but nematode parasitism also leads to tens of billions of dollars (USD) worth of livestock losses annually¹.'

Reviewer #1- Comment 4: Line 112 and elsewhere; agonism is a pharmacological concept referring to receptor activation, and 'agonizes' is not normally used in this context; 'stimulates' may be preferable;

Our Response: We have replaced agonism with stimulation or enhancement throughout the manuscript, including the title, which now reads: 'Nemertin is a Nematode-Selective Small Molecule Stimulator of Neurotransmitter Release'. We made ~23 changes to this effect.

Reviewer #1- Comment 5: Line 117: nemertin represents a potentially valuable scaffold (which is the alkyl phenylpiperidine moiety)

Our Response: We have modified the respective sentence to read, 'We conclude that the nemertin alkyl phenylpiperidine core scaffold is a novel nematode-selective nematocidal lead that may also improve the selectivity of broad-acting pesticides.'

Reviewer #1- Comment 6: Line 126: 'paragon' does not seem to be the right word here

Our Response: Thank you for catching that typo- it should have read, 'paradigm'- we have corrected the error.

Reviewer #1- Comment 7: Line 181: 'popular' should be replaced with 'commercially available'

Our Response: We have made the requested change.

Reviewer #1- Comment 8: not finding resistant mutants in *C. elegans* screens is a good thing but is not really proof that it will be slow to develop in the field (as I know the authors are aware; just need to be a bit more conservative in this case).

Our Response: The sentence in question is already pretty conservative, using the terms 'suggests' and 'may' as follows:

'These results suggest that: i) nemertin-1 does not share an MOA with canonical anthelmintics, ii) nemertin-1's MOA may not be limited to a single protein target, and, iii) genetic resistance to nemertin-1 may be difficult to achieve in the field.'

However, out of an abundance of caution, we have added the following parenthetical clause at the end of the third point: '(which is a hypothesis that remains to be tested)'

Reviewer #2-General Comments: The presented work on the discovery approach based on behavior of nematodes and the presentation of some molecules with anthelmintic action is a very important contribution and needs to be published to become available to a broader community. In particular, the approach on nematode behavior (motoric based) has been elegantly and convincingly laid out.

Our Response: We thank the reviewer for their kind words.

Reviewer #2- Comment 1: in significance statement, the statement's 2nd sentence is too absolute - not all existing anthelmintics are being "rendered ineffective because of the evolution of resistance". This depends largely on the hosts (more a problem in livestock less in companion animals) and on the nematode species. I recommend the addition of "many" at the start of this sentence, which still would highlight the apparent urgency.

Our Response: Communications Biology does not have a 'Significance Statement', so we have deleted the whole paragraph.

Reviewer #2- Comment 2: In the results and discussion section, the analysis of Nematode analogs assayed against free-living nematodes, parasitic nematodes, and non-target models (Supplementary Table 3; Supplementary Data 4) - considering the statement that nematode selective agonists are needed for animal health and crop production, the results on real parasitic nematodes (and not only on *C. elegans*) must be visible in the manuscript, also in an easier to read summary table.

Our Response: We thank the reviewer for this suggestion and feel that the spirit of this comment is justified. We addressed this request in three ways.

First, we have moved Supplementary Table 1 into the main display items (it is now Fig. 2).

Second, we have made Supplementary Table 3 a main display item (it is now Fig. 6).

Finally, we have better mapped out the correspondence between the main display items and where the corresponding detailed data can be found in the Supplementary Data Files (i.e. the supplementary excel tables) throughout the revision. For convenience, we list those files here:

Supplementary Data 1. Data for Fig. 1 and Fig. 2.

Supplementary Data 2. Data for Fig. 3

Supplementary Data 3. Data for Fig. 4

Supplementary Data 4. Data for Fig. 5

Supplementary Data 5. Data for Fig. 6

Supplementary Data 6. Data for Supplementary Fig. 2

Supplementary Data 7. Data for Supplementary Fig. 3

Supplementary Data 8. Data for Supplementary Fig. 4

With respect to the reviewer's last comment (providing an easier to read summary table), we apologize but we feel that we cannot further simplify the original Supplementary Table 3 (i.e. the SAR table with the different bioassays with targeted species and non-targeted models), which is now Fig. 6, because it is *already a summary* of a vast amount of data presented in the supplementary data files (Supplementary Data File 5). The new Fig. 6 is necessarily a large matrix that communicates the structures of the analogs, and the activity of those analogs in many bioassays. It is difficult for us to envision how simplifying this further could effectively

communicate the data without losing essential details. We have therefore elected to keep the table in its original form.

Reviewer #2- Comment 3: In the results and discussion section, interference with motoric action of nematodes is shown for the egg laying parameter. For animal pathogenic nematodes, motoric interference may be even more important for movement, due to necessity of nematodes to counteract against host gut peristaltic. The required concentrations may be different as is the case of macrocyclic lactones.

Our Response: We interpret that the reviewer may be making two important points here:

First, the reviewer is stating that disruption of egg-laying alone may not necessarily translate well to the disruption parasite movement. We agree that translation is never a certainty. However, we would like to remind the reviewer that egg-laying was only the first of several assays in our search for small molecule disruptors of nematode motor activity. The original Supplementary Table 1 (and now main display item Fig. 2) shows how we re-screened all molecules that disrupt *C. elegans* egg-laying in a locomotion (whole-body movement) assay. Additional *C. elegans* movement data is presented in Figure 3b and 3c (and elsewhere in the display items). Also, our structure-activity relationship (SAR) analysis presented in old Supplementary Table 3 (now main display item Fig. 6) reports the ability of nementin analogs to disrupt the movement of many parasites. We hope this is satisfactory.

A second point that we interpret the reviewer to be making is that the concentrations that are effective against one species may be different against a different species, and especially when that species is in its natural host or field environment, which again, we do not disagree with. To address this point, we have added the following sentence in the discussion:

The concentrations of nementin that are effective against *C. elegans* are likely to be different when used against parasites in the field; further experimentation is required to test this hypothesis.

Reviewer #2- Comment 4: For application against plant pathogenic nematodes it might be of value to add a beneficial - non pathogenic soil nematode to confirm the selective activity.

Our Response: Throughout the manuscript, including in the title, we use the term 'selectively' to indicate that nementin-1 has effects on nematodes without obviously effecting the non-nematode species tested. We think that this is an appropriate and obvious use of the term 'selective'. Nowhere in the manuscript is there a claim that nementin can incapacitate select species of nematodes. However, careful inspection of revised Fig. 6 (the SAR of the original Sup Table 3) shows a couple of nementin analogs (#6 and #10) that demonstrate poor activity against the free-living species, but obvious activity against some parasitic nematodes. Given that intra-Nematoda selectivity is not a focus of this work, we will leave it up to the reader to determine whether they see potential value in pursuing nementin as an intra-Nematoda selective nematicide.

Reviewer #2- Comment 5: To validate and possibly even more substantiate the value of the presented approach, the effect of nementin on ML- and levamisole-resistant nematodes could be presented. If available, activity on drug resistant nematodes should be pointed out, if not, those needs to be discussed.

Our Response: Supplementary Figure 2 shows exactly this data for *C. elegans* mutants that resist MLs (ivermectin) and levamisole. *C. elegans* mutants that resist these molecules and other anthelmintics/nematicides remain sensitive to nementin. That said, we believe the reviewer may be referring to nematode parasites from the field that resist anthelmintics/nematicides, which we have not tested. To address this point, we added the following sentence in the discussion:

In addition, it is currently unclear whether nementin can incapacitate drug-resistant parasitic strains, despite showing that drug-resistant *C. elegans* mutants remain sensitive to nementin (Supplementary Fig. 2).

Reviewer #2- Comment 6: Details: pipeline or approach? Pipeline (eg in introduction) in parasiticide discovery is defined as a summary of available molecules at various stages in the discovery process. In here the authors present a very valuable specific approach for discovery and on top present various molecules but focus on one, so “approach” maybe more applicable.

Our Response: The manuscript describes a successive series of screens that leads us to chose one molecule for detailed investigation. Our first screen is that of 486 wactive molecules for those that induce egg-laying defects, resulting in 58 egg-laying modulators. Those egg-laying modulators are then put through a second screen for locomotory defects, yielding 26 compounds. Based on known properties of these 26 molecules, 7 are chosen for further investigation. Additional bioassays show that one molecule, wact-55 (aka nementin), arguably performed the best in these assays. Given this, we feel that the term ‘pipeline’ is appropriately used to describe the workflow.

Reviewer #2- Comment 7: name -nementin: is the name applicable to the WHO process of naming novel anthelmintics? If the class of molecules would become valuable for application, naming might be different.

Our Response: From the WHO document titled ‘Guidance on the use of International non-proprietary names (INNs) for pharmaceutical substances (2017)’ INN naming follows a defined convention: ‘Usually, an INN consists of a random, fantasy prefix and a common stem; substances belonging to a group of pharmacologically related substances show their relationship by the use of a common stem’ (page 11). Annex 3 of this guideline titled ‘List of common stems used in the selection of INNs’ notes four stems used for anthelmintics: *-antel*, *-bendazole*, *-fos*, *-vos*; and two stems for antiparasitics: *-ectin*, *oxanide*; in each instance, these stems refer to distinct anthelmintic/antiparasitic chemical structures (and associated mechanisms). Our proposed stem ‘*-entin*’ or even shorter ‘*-tin*’ has not been used for any other INN named compound. Given we have shown that nementin likely acts through a novel mechanism and is chemically distinct from commercial anthelmintics, we feel that if nementin is eventually developed into an anthelmintic that warrants an INN name designation, nementin would be an appropriate candidate name for submission.

Reviewer #2- Summary: I would recommend to consider some of my comments (for the authors) for some minor revisions, particularly adding results on true parasitic nematodes in an accessible and readable form. Overall, the MS should definitely be published with some minor revisions.

Reviewer #3-General Comments: The increasing global drug resistant problem in parasitic nematodes which cause huge human and animal diseases and enormous losses of crop production makes the new drug discovery and vaccine development an urgent task. This manuscript identified a compound named nementin by using a pipeline exploiting multiple motor outputs of the model nematode *C. elegans*. It is found that nementin-1 can induce worm's convulsions and paralysis by agonizing neuronal dense core vesicle release and cholinergic signaling. By consequence, nementin synergistically enhances the activity of nematode's AChEIs, implicating the potential development of nementin as an environment-friendly nematicide. The aim of this work is cleanly formulated, the methods are presented in detail and the results are clearly described. I would suggest to accept for publication after a minor revision.

Our Response: We thank the reviewer for their kind words.

Reviewer #3- Comment 1: Line 218, delete “()”.

Our Response: Thanks for catching that. We have corrected the error.

Reviewer #3- Comment 2: Lines 225, “4” should be changed into “3”.

Our Response: Thanks for catching that. We have corrected the error.

Reviewer #3- Comment 3: Line 275, delete “.” After “a”. “UNC-1” should be read as “UNC-13”.

Our Response: Thanks for catching that. We have corrected the error.

Reviewer #3- Comment 4: Line 620, “(d-f)” should be changed into “(d, f, h)”. Line 622, “(g-i)” should be changed into “(e, g, i)”.

Our Response: Thanks for catching that. We have corrected the error.

REVIEWERS' COMMENTS:

Reviewer #1 (Remarks to the Author):

The authors have responded constructively and positively to the minor concerns raised by me and the other reviewers. The changes made as a consequence of this review process have fully resolved the concerns and the manuscript should now be processed for publication.

Reviewer #2 (Remarks to the Author):

The reviewer thanks the authors for their considerations and adaptations of their valuable manuscript.

All but one point is agreeable: The term "pipeline" is in my view not appropriate to describe the approach or flowchart or sequence series of screens, because it is used as a term for the available drug candidates in a system at a given time. THus, "pipeline" may be misleading by some readers. However, this may only be a personable view and is certainly not critical at all for this excellent paper.

Reviewer #3 (Remarks to the Author):

The authors have revised the manuscript. I am satisfied with the revision. I recommend to accept for publication to this journal.